# Mixed alkali-ion transport and storage in atomic-disordered honeycomb layered NaKNi$_2$TeO$_6$

Titus Masese [1,2✉], Yoshinobu Miyazaki[3✉], Josef Rizell[1,4], Godwill Mbiti Kanyolo [5✉], Chih-Yao Chen[2], Hiroki Ubukata[6], Keigo Kubota [2], Kartik Sau [1,7], Tamio Ikeshoji[7], Zhen-Dong Huang[8✉], Kazuki Yoshii [1], Teruo Takahashi[3], Miyu Ito[3], Hiroshi Senoh [1], Jinkwang Hwang[9], Abbas Alshehabi[10], Kazuhiko Matsumoto [2,9], Toshiyuki Matsunaga[11], Kotaro Fujii [12], Masatomo Yashima [12], Masahiro Shikano [1], Cédric Tassel[6], Hiroshi Kageyama [6], Yoshiharu Uchimoto [11], Rika Hagiwara [2,9] & Tomohiro Saito[3✉]

Honeycomb layered oxides constitute an emerging class of materials that show interesting physicochemical and electrochemical properties. However, the development of these materials is still limited. Here, we report the combined use of alkali atoms (Na and K) to produce a mixed-alkali honeycomb layered oxide material, namely, NaKNi$_2$TeO$_6$. Via transmission electron microscopy measurements, we reveal the local atomic structural disorders characterised by aperiodic stacking and incoherency in the alternating arrangement of Na and K atoms. We also investigate the possibility of mixed electrochemical transport and storage of Na$^+$ and K$^+$ ions in NaKNi$_2$TeO$_6$. In particular, we report an average discharge cell voltage of about 4 V and a specific capacity of around 80 mAh g$^{-1}$ at low specific currents (i.e., < 10 mA g$^{-1}$) when a NaKNi$_2$TeO$_6$-based positive electrode is combined with a room-temperature NaK liquid alloy negative electrode using an ionic liquid-based electrolyte solution. These results represent a step towards the use of tailored cathode active materials for "dendrite-free" electrochemical energy storage systems exploiting room-temperature liquid alkali metal alloy materials.

[1] Research Institute of Electrochemical Energy, National Institute of Advanced Industrial Science and Technology (AIST), Osaka, Japan. [2] AIST-Kyoto University Chemical Energy Materials Open Innovation Laboratory (ChEM-OIL), Kyoto, Japan. [3] Tsukuba Laboratory, Technical Solution Headquarters, Sumika Chemical Analysis Service (SCAS), Ltd., Tsukuba, Japan. [4] Department of Physics, Chalmers University of Technology, Göteborg, Sweden. [5] Department of Engineering Science, The University of Electro-Communications, Tokyo, Japan. [6] Department of Energy and Hydrocarbon Chemistry, Graduate School of Engineering, Kyoto University, Kyoto, Japan. [7] Mathematics for Advanced Materials—Open Innovation Laboratory (MathAM-OIL), National Institute of Advanced Industrial Science and Technology (AIST), c/o Advanced Institute of Material Research (AIMR), Tohoku University, Sendai, Japan. [8] Key Laboratory for Organic Electronics and Information Displays and Institute of Advanced Materials (IAM), Nanjing University of Posts and Telecommunications (NUPT), Nanjing, China. [9] Graduate School of Energy Science, Kyoto University, Kyoto, Japan. [10] Department of Industrial Engineering, National Institute of Technology (KOSEN), Ibaraki, Japan. [11] Graduate School of Human and Environmental Studies, Kyoto University, Kyoto, Japan. [12] Department of Chemistry, School of Science, Tokyo Institute of Technology, Tokyo, Japan. ✉email: titus.masese@aist.go.jp; yoshinobu.miyazaki@scas.co.jp; gmkanyolo@mail.uec.jp; iamzdhuang@njupt.edu.cn; tomohiro.saito@scas.co.jp

Honeycomb layered oxides are a family of lamellar-structured nanomaterials characterised mainly by alkali or coinage-metal atoms interleaved between sheets of transition metal atoms aligned in a honeycomb formation. This emerging class has been gaining momentous interest as a result of a variety of appealing properties innate to their structural framework[1–10]. The alkali- or coinage-metal atoms manifest weak interlayer bonds that engender an abundance of unoccupied sites that induce excellent ionic conductivities. This allows for facile reinsertion and extraction of alkali ions between the transition metal sheets, thus making them ideal cathode active material candidates for rechargeable alkali-based batteries[1,2,11–17]. Furthermore, the sandwiching of non-magnetic atoms between a hexagonal sublattice comprising magnetic atoms results in pseudo-two-dimensional magnetic structures that have the potential to achieve exotic magnetic states with varied applications in fields such as quantum computing and solid-state physics[1,10].

Most honeycomb layered oxides encompass compositions; $A_2M_2DO_6$, $A_3M_2DO_6$ or $A_4MDO_6$ where $A$ is an alkali- or coinage-metal atom ($A$ = Li, Na, K, Cu, Ag, …), $M$ is a transition metal atom ($M$ = Ni, Co, Mg, Zn, Mn, Fe, Cr, …) and $D$ is a highly valent ion like Te, Sb, Bi or W[1]. In these compositions, $A$ atoms are sandwiched between slabs comprising $M$ atoms surrounded by $D$ atoms in a hexagonal formation. Depending on the atomic size of the $A$ atom, the resulting lamellar structures manifest different sequential arrangements (hereupon referred to as stacking orders). Until now, only a handful of honeycomb layered oxides adopting T2-, O1-, O3- and P2-type (in Hagenmuller-Delmas' notation) stacking orders have been identified, whereby 'T', 'O' and 'P' denote the tetrahedral, octahedral or prismatic coordination of oxygen with the $A$ atoms, whilst the ensuing digit corresponds to the number of repeating transition metal layers in the unit cell[18]. In order to further expand the scope of known honeycomb layered oxides and capitalise on their full potential, it is imperative to not only explore hitherto uncharted territories of their compositional space but also scrutinise their emergent stacking orders.

Amongst other classes of layered transition metal oxides, fascinating structures have been developed through the mixing of two different alkali species to formulate $A_xA'_yMO_2$ compositions. For instance, in the commonly studied $Li_xNa_yCoO_2$ layered cobaltate, the intermixing of similar amounts of Na and Li results in unique configurations of Na and Li atoms within the different layers giving rise to versatile stacking structures ranging from OP4 to OPP9 stacking sequences[19,20]. This structural versatility facilitates the development of various crystal structures with the potential to host different functionalities. Materials such as $Li_{0.48}Na_{0.35}CoO_2$ have been found to exhibit a large thermoelectric power (thermopower) at room temperature, surpassing that of either of its parent materials, $Na_yCoO_2$ and $Li_xCoO_2$[21]. Furthermore, the unique OP4 stacking sequence has found great utility in battery application as it allows $Li_xNa_yCoO_2$ to be utilised as a precursor in ion-exchange synthesis to create a new polymorph electrode material $LiCoO_2$ with O4-stacking[22,23]. Although the intermixing of alkali ions appears to be a judicious approach to the next level development of honeycomb layered oxides, as far as we can tell, only one report on a set of metastable antimonates $Li_{3-x}Na_xNi_2SbO_6$ has been published on the topic[24].

Herein, we investigate a novel composition of $Na_{2-x}K_xNi_2TeO_6$ honeycomb layered oxides. Furthermore, we unravel the structure of the new mixed-alkali ion layered oxide $NaKNi_2TeO_6$ using atomic-resolution scanning transmission electron microscopy (STEM). Visualised for the *first time*, the local atomic structure reveals a unique and aperiodic stacking sequence in the layered mixed-alkali ion compound. We investigate the ability of $NaKNi_2TeO_6$ to store $Na^+$ or $K^+$ ions when coupled with metallic sodium or potassium electrodes. $NaKNi_2TeO_6$ is also tested in combination with a liquid NaK alloy anode using an ionic liquid-based electrolyte solution for room-temperature dendrite-free rechargeable dual-cation cell.

## Results

In this study, we utilised $Na_2Ni_2TeO_6$ and $K_2Ni_2TeO_6$ as the parent materials for the creation of a novel stable mixed-alkali ion phase. The P2-type stacking (crystallising in the centrosymmetric $P6_3/mcm$ hexagonal space group) exhibited by both parent materials ($Na_2Ni_2TeO_6$ and $K_2Ni_2TeO_6$) is explicitly illustrated in Fig. 1a, together with possible structural models for the resulting mixed compounds. Mixed-alkali ion honeycomb layered oxides adopting the composition of $Na_{2-x}K_xNi_2TeO_6$ ($0 \leq x \leq 2$) were synthesised via a high-temperature solid-state synthesis route described in the "Methods" section. A preliminary characterisation of the average crystal structures of the as-synthesised (pristine) materials was carried out using powder X-ray diffraction (XRD) (as shown in Fig. 1b). When the smaller Na atoms were replaced with larger K atoms (viz., increasing $x$ from 0 to 2), a stepwise shift comprised of several diffraction peaks was observed. Particularly, two discrete shifts are seen as $00l$ peaks of the $Na_{2-x}K_xNi_2TeO_6$ as shown in Fig. 1c. Since the diffraction angles of the $00l$ peaks are inversely proportional to the distance between adjacent transition metal slabs (hereafter referred to as the interlayer distance), each position corresponds to a phase with a different interlayer distance. Two of the phases have interlayer distances closely resembling the parent materials $Na_2Ni_2TeO_6$ and $K_2Ni_2TeO_6$, whilst the last phase has an intermediate interlayer distance, indicating the existence of both Na and K atoms in this phase.

In order to quantify and better illustrate how the lattice parameters of the phases present in $Na_{2-x}K_xNi_2TeO_6$ change with varying amounts of Na and K, profile fitting (Le Bail fit) of the XRD patterns was subsequently carried out. The average lattice parameters as deduced from the fit, are provided in the Supplementary Information (Supplementary Fig. 1). $Na_{2-x}K_xNi_2TeO_6$ compositions where $0.2 \leq x \leq 1.8$ are treated as two-phase mixtures, since the $00l$ Bragg peaks split into two separate peaks in these samples (as shown in Fig. 1c). When K content ($x$, in $Na_{2-x}K_xNi_2TeO_6$) is increased with $Na_2Ni_2TeO_6$ as the starting material, the relative intensity of the peaks corresponding to the $Na_2Ni_2TeO_6$ phase decrease in favour of the new intermediate phase. It should be noted that when an equimolar ratio of Na and K is reached (i.e., $NaKNi_2TeO_6$), the $Na_2Ni_2TeO_6$ peaks disappear and new peaks emerge (as shown in Fig. 1d). With further increase in the K content, the relative intensities of the peaks corresponding to the intermediate phase decreases and some other peaks appear accordingly until pure $K_2Ni_2TeO_6$ is attained.

To reiterate, amongst the $Na_{2-x}K_xNi_2TeO_6$ diffraction patterns, peaks previously not found in the parent materials were observed in the mixed-alkali compound. As illustrated by Fig. 1d, a peak located at lower diffraction angles emerges in the intermediate compositions despite being forbidden in the $P6_3/mcm$ hexagonal space group used to index both $Na_2Ni_2TeO_6$ and $K_2Ni_2TeO_6$. Similar observations have been noted on a previous report on $Li_{3-x}Na_xNi_2SbO_6$ whereby disparate peaks emerged when Li and Na atoms were separated in different layers[24], suggesting new cationic ordering in these materials. As such, the disappearance of the 102 Bragg peak and the emergence of a new set of peaks in close proximity underline the structural changes occurring in this intermediate phase.

The tendency of these mixed-alkali ion compositions to separate into two-phase mixtures can be rationalised by the large

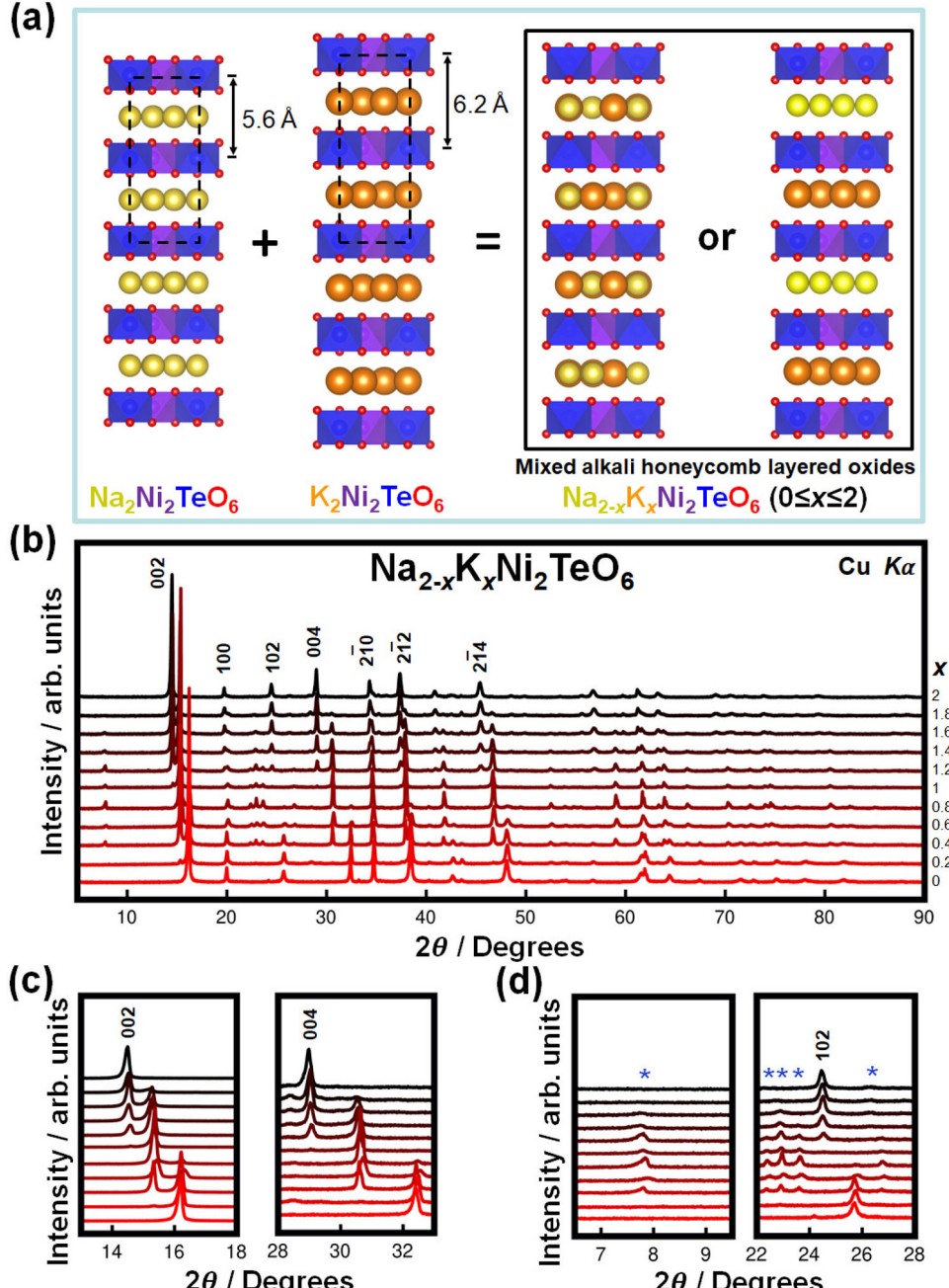

**Fig. 1 Structural characterisation of Na$_{2-x}$K$_x$Ni$_2$TeO$_6$ (0 ≤ x ≤ 2). a** Schematic illustration of possible structural models in mixed-alkali atom honeycomb layered oxides adopting the composition Na$_{2-x}$K$_x$Ni$_2$TeO$_6$ (0 ≤ x ≤ 2). In the isostructural $A_2$Ni$_2$TeO$_6$ ($A$ = Na, K) compounds, Na atoms (in yellow) or K atoms (in orange) are sandwiched between layers or slabs consisting exclusively of TeO$_6$ (blue) and NiO$_6$ (purple) octahedra. Black dashed lines denote the unit cell. Owing to the larger Shannon–Prewitt ionic radius of K$^+$ (1.38 Å) compared to Na$^+$ (1.02 Å), the interlayer distance of K$_2$Ni$_2$TeO$_6$ is significantly larger than that of Na$_2$Ni$_2$TeO$_6$. Various reasonable structural models can be hypothesised for the new series of compounds adopting the composition Na$_{2-x}$K$_x$Ni$_2$TeO$_6$ (0 ≤ x ≤ 2). Here, models where Na and K atoms are either mixed within the same layers or separated into different layers are shown. **b** XRD patterns of as-synthesised Na$_{2-x}$K$_x$Ni$_2$TeO$_6$ (0 ≤ x ≤ 2), showing a stepwise shift of Bragg peaks towards lower diffraction angles with the substitution of Na with K. Bragg peaks for K$_2$Ni$_2$TeO$_6$ that are indexed in the hexagonal $P6_3/mcm$ space group are shown in black. Results of the elemental composition analyses of the prime intermediate compositions are furnished in the Supplementary Information (Supplementary Table 1 and Supplementary Figs. 2–8). **c** A stepwise shift of the 002 and 004 Bragg peaks that clearly hallmark the increase in the interlayer distance with K atom substitution. The three distinct peak positions correspond to three phases with different interlayer distances (along the $c$-axis), with the intermediate peak position signifying the formation of a new phase that is different from the end members Na$_2$Ni$_2$TeO$_6$ and K$_2$Ni$_2$TeO$_6$. **d** Emergence of new Bragg peaks in the intermediate compositions (0 ≤ x ≤ 2), which are not allowed in the $P6_3/mcm$ hexagonal space group. These new peaks are marked by asterisks and are indicative of a symmetry change that may entail the formation of superstructures.

difference in the Shannon–Prewitt radii of $Na^+$ and $K^+$ ions[25], making it difficult to form a solid-solution compound from a mixture of $Na_2Ni_2TeO_6$ and $K_2Ni_2TeO_6$. Similar behaviour has also been observed amongst the layered nickelates $Na_xLi_{1-x}NiO_2$, where three different polymorphs with intermediate two-phase regions form depending on the stochiometric Li/Na ratio, presumably as a result of the large difference between the ionic radii of Li and Na[26]. To further elucidate the manner of alkali atom arrangement upon successful intermixing of Na and K and the corresponding structural changes previously indicated by the XRD patterns (Fig. 1d), crystal structural analyses were employed on $NaKNi_2TeO_6$ whose composition is closest to a phase-pure sample. It is prudent to mention here that the elemental composition and thermal stability of $NaKNi_2TeO_6$ were ascertained by inductively coupled plasma measurements and thermal gravimetric analyses, respectively (Supplementary Table 2, Supplementary Figs. 9, 10).

Synchrotron XRD measurements of $NaKNi_2TeO_6$ were performed for in-depth structural analyses, and its refinement result is shown in Supplementary Fig. 11. We performed a Le Bail profile fitting using a hexagonal unit cell, yielding the lattice parameters of $a = 5.2258(1)$ Å and $c = 11.7875(6)$ Å and the reliability factors of $R_{wp} = 8.96\%$, $R_p = 6.27\%$ and goodness-of-fit (GOF) = 4.12. We initially considered the reported structure of $Na_{1.97}Ni_2TeO_6$ in the $P6_3/mcm$ space group[11], but disregarded this hexagonal space group because of the presence of the 001 and 003 reflections at ~3.01° and 9.05°. We also noticed that several peaks could not be fitted well with the initial structure considered. For example, the 100 and 003 reflections (Supplementary Fig. 11b) were tailed and shifted towards higher angles, as previously observed in several layered oxides containing stacking faults (e.g., $Li_2NiO_3$[27]). Preliminarily, we adopted several hexagonal models (some of which are shown in Supplementary Fig. 12), in which Na and K atoms are alternately arranged in a honeycomb layered framework. To ascertain information about the structure of $NaKNi_2TeO_6$, especially the stacking arrangement, atomic-resolution imaging of pristine $NaKNi_2TeO_6$ was conducted along several zone axes. Information on sample preparation, measurement protocols, and caveats undertaken are explicated in the "Methods" section.

The atomic-resolution imaging was accomplished using an aberration-corrected scanning transmission electron microscopy (STEM). From the [001] zone axis, the characteristic honeycomb arrangement of transition metal atoms can be observed. Fig. 2a shows a high-angle annular dark-field (HAADF)-STEM image of the parent $K_2Ni_2TeO_6$. As the intensity is approximately proportional to the square of the atomic number $(Z)$[28–30], the honeycomb arrangement of heavier elements Te $(Z = 52)$ and Ni $(Z = 28)$ should be explicitly visualised through spots of varying intensities. This is evident in the analogous $K_2Ni_2TeO_6$ (see Supplementary Fig. 13), where the red spots indicate columns of Ni atoms whilst the bright yellow spots correspond to columns of Te atoms. As such, it should be possible to acquire an analogous image for $NaKNi_2TeO_6$ if its honeycomb slab structure is similar to that of $K_2Ni_2TeO_6$. However, Fig. 2a shows that all atomic sites in the image share the same intensity, indicating "overlapping" of Ni and Te atoms in adjacent layers. It was also elusive to discern lighter (lower atomic mass) elements such as K and Na in the corresponding annular bright-field (ABF)-STEM images (shown in Fig. 2b).

To clarify the structural changes responsible for the overlap of Ni and Te atoms observed in $NaKNi_2TeO_6$, the crystallites were examined from different directions (zone axes). A view from the [100] direction reveals the lamellar nature of the structure—in the HAADF-STEM image (Fig. 2c), Te and Ni atoms correspond to a set of bright planes. This assignment was further confirmed by

augmenting the STEM images with energy dispersive X-ray spectroscopy (EDX), as shown in Supplementary Fig. 14. Elemental mapping using EDX also reveals that Na and K atoms are sandwiched between these Ni/Te slabs. Furthermore, the STEM images reveal that Na and K atoms are separated into different layers, instead of being randomly mixed within the same layers (Fig. 2d–f).

Further analyses of atomic-resolution STEM images from the [100] direction allow the stacking order in $NaKNi_2TeO_6$ to be characterised. Notably, the shift between adjacent Ni/Te slabs is contingent on whether the interlayer space is occupied by Na or K atoms (Fig. 2g). No shift is observed when Ni/Te slabs are separated by a K layer. The alternating arrangement of Na and K atom layers was observed also when viewed along the [1$\bar{1}$0] zone axis, as shown in Supplementary Fig. 15. Moreover, the transition metal slabs sandwiching Na atoms are shifted by a period of 1/3 along or parallel to adjacent slabs. This stacking shift explains the observation of Ni and Te spots with approximately equal intensity when observed from the [001] direction (Fig. 2a, b), presumably owing to their overlapping. The direction of the slab shifts seems to be random lacking any periodicity, which explains the difficulties encountered in the structural and profile analyses of the powder diffraction patterns.

Attempts to determine whether the structure is truly random or aperiodic, albeit ordered to some degree, proved elusive. From an enlarged view of the atomic-scale HAADF-STEM mapping (Fig. 2h), it is also evident that the interlayer distance depends on the alkali atom species sandwiched between adjacent Ni/Te layers. The Ni/Te layers with Na atoms are separated by 0.55 nm (5.5 Å) whilst the interlayer distance for the layers with K is 0.62 nm (6.2 Å). It is worth highlighting that these interlayer distances attained closely resemble those of the parent compounds $Na_2Ni_2TeO_6$ and $K_2Ni_2TeO_6$ (Fig. 1a). The intensity line profiles (shown in Fig. 2i) quantitatively illustrate the alternating interlayer distances of Na and K atoms. Exclusively relying on the XRD patterns only yields the average of these interlayer spacings/distances (Supplementary Fig. 1), accentuating the efficacy of using TEM to obtain important structural information.

Although the atomic structure of $NaKNi_2TeO_6$ exhibits significant aperiodicity, a partial structural model containing the slab shift can still be constructed based on the TEM analysis. Fig. 3a illustrates an atomistic model viewed from the [100] direction, where the atomic coordinates based on $K_2Ni_2TeO_6$ structure obtained from XRD analysis were used. This model can be superimposed on both HAADF-STEM and ABF-STEM images, to confirm the accuracy of the positions of both heavier and lighter elements (Fig. 3b, c). Comparison between the average structure model and the kinematically simulated selected area electron diffraction (SAED) pattern further supports the validity of the obtained model (Fig. 3d, e). The overall appearance of the simulation based on our model seems similar to the ones observed in the experiment, indicating that this partial model represents the average structure relatively well. However, the deviation of the peaks from the periodic arrangement and the presence of streaks along the [001] direction in the experimental SAED patterns (Fig. 3d) attest to the aforementioned aperiodic stacking sequence.

To fully capture the atomistic model of $NaKNi_2TeO_6$, analyses along the [1$\bar{1}$0] zone axes are complementary, as shown in the partial structure model (Fig. 3f). Superimposing the model on both ABF-STEM and HAADF-STEM images yields excellent resemblance, thus confirming the accuracy of the alkali atom sites. In addition, the positions of Ni and Te atoms can be clearly distinguished in the HAADF-STEM image (Fig. 3g). It should be noted that the shift of the metal slabs that was observed for all Na layers in the [100] direction was not observed in the [1$\bar{1}$0]

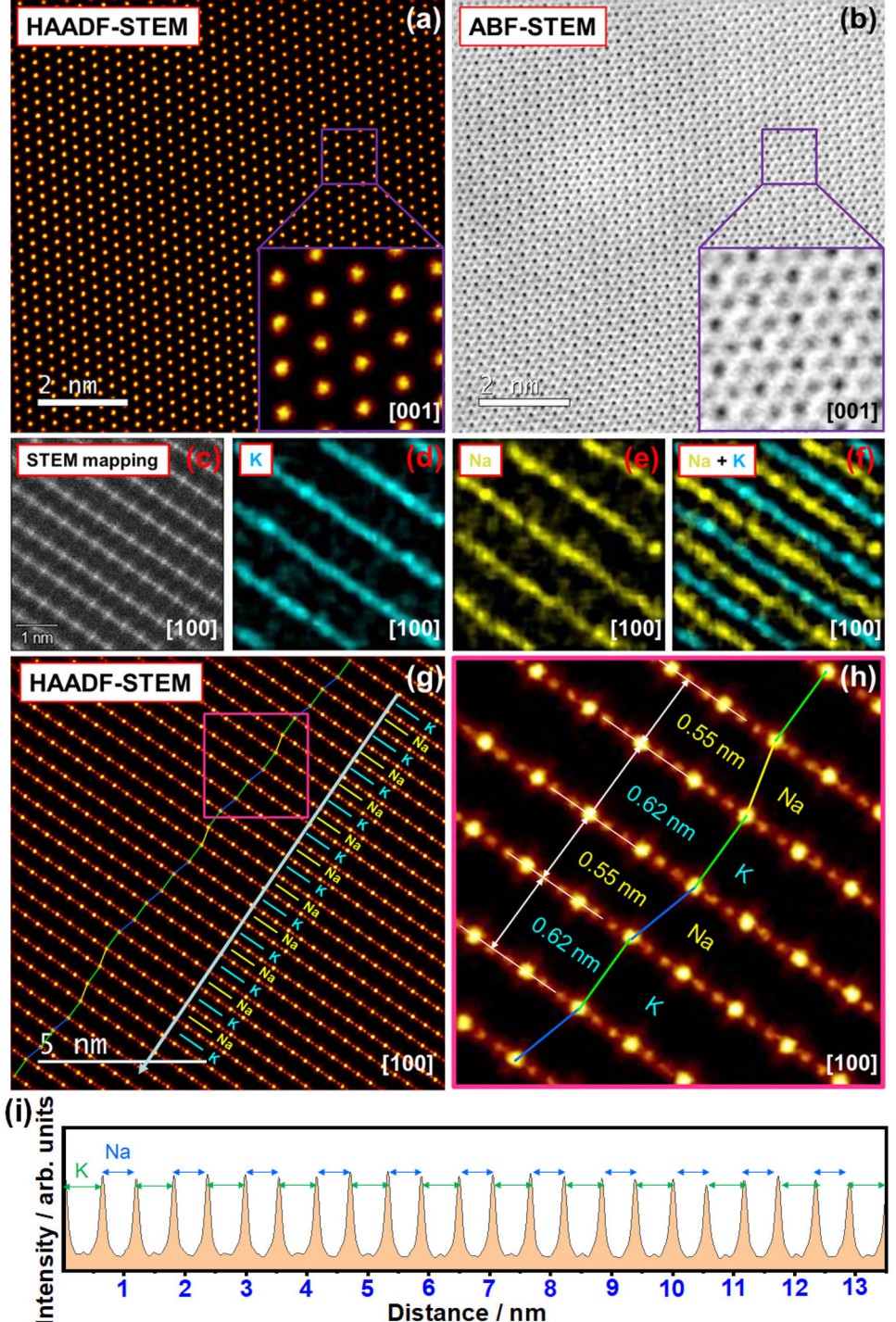

**Fig. 2 Arrangement of atoms in NaKNi$_2$TeO$_6$, viewed along the [001] and [100] directions. a** High-angle annular dark-field (HAADF) scanning transmission electron microscopy (STEM) images taken along the [001] zone axis. Overlap of Te and Ni atom positions in the adjacent layers along the *c*-axis result in spots with uniform intensity (when viewed in the [001] zone axis). **b** Corresponding annular bright-field (ABF)-STEM image, where also lighter elements (Na, K and O) can be visualised. **c** HAADF-STEM image of NaKNi$_2$TeO$_6$ taken along the [100] zone axis, revealing bright planes corresponding to the layers comprising Te and Ni, as further explicated in the corresponding energy dispersive X-ray (EDX) imaging (see Supplementary Information (Supplementary Fig. 14) section). **d–f** STEM-EDX mapping of the area shown in (**a**), where the colours explicitly visualise the distribution of Na and K atoms. The EDX maps show that the layers occupied by Na alternate with those of K. **g** HAADF-STEM image illustrating the unique stacking sequence in NaKNi$_2$TeO$_6$. In the layers where K atoms occupy the interlayer space, Te/Ni slabs are not shifted with respect to each other (marked by a green line). However, for layers where Na atoms reside, ±1/3 shifts of the Te/Ni slabs are observed. The yellow and blue lines show shifts in different directions. Note the aperiodicity in the stacking sequence. **h** Enlarged view of the domain highlighted in (**e**), showing that the interlayer distance is contingent upon the alkali atom species (Na or K) sandwiched between the Te/Ni layers. **i** Line profile showing alternating interlayer distances occupied by K and Na, as shown by an arrow line in (**g**).

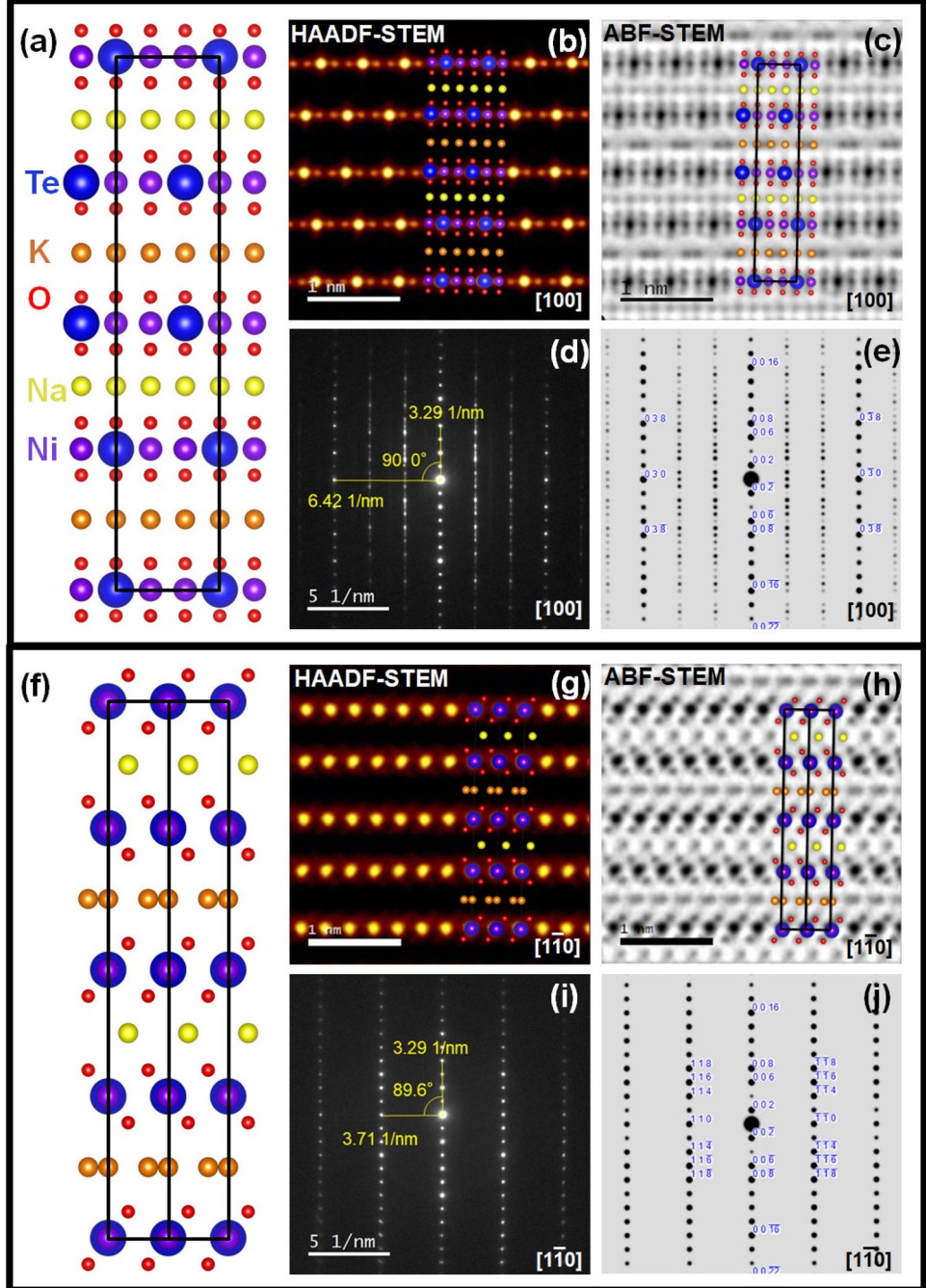

**Fig. 3 Aperiodic atomic structure of NaKNi$_2$TeO$_6$ along the [100] and the [1$\bar{1}$0] zone axes.** Aperiodic atomic structure of NaKNi$_2$TeO$_6$ along the [100] and the [1$\bar{1}$0] zone axes. **a** Atomistic model of the average aperiodic structure of NaKNi$_2$TeO$_6$ acquired based on STEM analyses along the [100] zone axis. Black lines show the partial unit cell. **b** Superimposition of the model on a HAADF-STEM image, showing an excellent overlap between the positions of the Ni and Te atoms in the model that is in accord with the intensity distribution of the atom spots observed in the image. **c** Superimposition of the model on an annular bright-field (ABF) image, affirming the atomic positions of Na, K and O. **d** Selected area electron diffraction (SAED) patterns taken along the [100] zone axis revealing spot shifts and streaks that are suggestive of the existence of aperiodicity. **e** Corresponding kinematic simulations based on the structural model shown in (**a**), showing agreement with the experimental results in (**d**). **f** Atomistic model of NaKNi$_2$TeO$_6$ acquired based on STEM analyses along the [1$\bar{1}$0] zone axis. Black lines show the partial unit cell. Note the difference in the arrangement of atoms between Na (yellow) and K (orange) crystallographic sites. **g** Superimposition of the model on a HAADF-STEM image, affirming the atomic positions of Ni and Te. **h** Superimposition of the model on an ABF-STEM image, also showing an excellent overlap between the positions of the Na, K and O atoms in the model that is in accord with the intensity distribution of the atom spots observed in the image. **i** SAED patterns taken along the [1$\bar{1}$0] zone axis and **j** the corresponding kinematic simulations, validating good agreement with the experimental results shown in (**d**).

direction. Thus, the slab shift can be described as [±2/3 ± 1/3 0], corresponding to the swapping of Ni and Te site, which explains why Ni and Te sites were indistinguishable when observed in the [001] direction. Correspondingly, in the ABF-STEM super-imposition, the Na and K atoms are seen as bright grey spots between the darker Te/Ni-atom planes (Fig. 3h, showing a clear difference between Na and K atom sites. The prismatic coordination of oxygen is clearly seen in the ABF-STEM images in both the Na and K layers. However, Na-atom sites are equidistantly spaced, whilst two adjacent K atom sites are grouped together.

Kinematically simulated SAED patterns in the [1$\bar{1}$0] direction generated based on the model agree well with the experimentally determined diffraction pattern (Fig. 3i, j). No streaks nor peak deviations from periodic arrangement are observed in the diffraction pattern from the [1$\bar{1}$0] zone axes (Fig. 3i), as opposed to the [100] direction (Fig. 3d). It validates that the [1$\bar{1}$0] projected structure is completely periodic since the shift of Ni/Te layers possessing the aperiodicity occurs along this direction. In an attempt to reconcile the refinement results based on the synchrotron XRD data of $NaKNi_2TeO_6$, a comparison of the experimental SAED patterns with the structural model indexed in $P\bar{6}2c$ hexagonal space group was performed (Supplementary Fig. 16), revealing $P\bar{6}2c$ hexagonal space group as appropriate to index the XRD pattern of $NaKNi_2TeO_6$. Rietveld refinement results of the synchrotron XRD and neutron diffraction patterns of $NaKNi_2TeO_6$ are furnished as Supplementary Information (Supplementary Figs. 17, 18), with the refined parameters in Supplementary Tables 3, 4. However, several issues with peak intensities and asymmetric peak profiles, which arise from the stacking disorder of the transition metal slab layers are apparent in both the XRD and neutron diffraction data that warranted further detailed structural analyses using TEM.

Structural intricacies of the mixed-alkali honeycomb layered oxide framework of $NaKNi_2TeO_6$ were further divulged by atomic-resolution STEM images (Supplementary Fig. 19), which reveal the existence of stacking disorders/faults wherein ±1/3 shifts of the Te / Ni slabs are observed. To further quantitatively scrutinise the nature of the stacking faults innate in $NaKNi_2TeO_6$ as revealed by STEM (Supplementary Fig. 19), the FAULTS program was employed[31]. The atomic structural model indexed in the $P\bar{6}2c$ hexagonal space group was used as the initial model to perform the analyses of the stacking faults (as shown in Supplementary Fig. 20). Various shift vectors were adopted to describe the stacking faults along the $c$ plane, amongst which, shift vectors of [−1/3, −1/3, 0], [1/3, 0, 0] and [0, 1/3, 0] were found to be dominant with stacking probabilities of 7.5%, 6.3% and 4.1%, respectively. Taking all the results altogether, $NaKNi_2TeO_6$ possesses a high degree of stacking faults and we anticipate further study on the defect chemistry and physics of this class of honeycomb layered oxide materials in another scope of work.

## Discussion

A detailed characterisation of the crystal structure of the mixed-alkali honeycomb layered oxide ($NaKNi_2TeO_6$) was achieved using aberration-corrected STEM. To the best of our knowledge, studies on the local atomic structure of a mixed-alkali ion layered oxide with a similar structure have not been previously reported, making information on this class of materials obscure and underutilised. In the course of this study, a prominent aspect that emerges is the difference between Na and K atom sites evident when $NaKNi_2TeO_6$ crystals are viewed along the [1$\bar{1}$0] and [100] zone axes. We find that, Na atoms are distributed in sites that assume triangular patterns (Fig. 4a) whilst the K atoms reside in sites arranged in honeycomb formations (Fig. 4b). Although beyond the limits of present experiments, these atomic

configurations and crystallographic occupations can be used to predict emergent properties of such materials as well as the electrodynamics of the alkali or coinage atoms within the realm of their electromagnetic behaviour, electrochemistry, quantum phenomena etc[32]. For instance, the atomic arrangements of the atoms given in Fig. 4a, b are intricately linked to crystalline entropy considerations. This, in turn, can be envisaged to impact the electrodynamics of alkali ions in the material.

In particular, the free energy of the alkali atom layer is expected to be minimised (or equivalently, the entropy maximised) to achieve stability of the crystal especially when the Na or K atoms are arranged in a honeycomb fashion. This follows from the honeycomb conjecture, which states that the honeycomb lattice is the most efficient way of packing any two-dimensional (2D) surface with equal size unit cells of maximum area and minimum perimeter[33]. Thus, taking the free energy $F$ to scale with the perimeter of the unit cells (honeycomb, triangular etc.), and the entropy to scale with the area, A, the lattice exhibited by K atoms in Fig. 4b satisfies this conjecture by maintaining its honeycomb pattern whilst Na atoms in Fig. 4a do not.

For the sake of rigour, we make a straightforward approximation for the free energy, $F$ using the thermodynamics formula, $F = U - k_B T \ln KA$, where $K$ is a constant related to the geometry of the surface (Gaussian curvature)[32], $S = k_B \ln KA$ is the entropy contribution to the free energy and $U \simeq -E_a$ is the internal potential energy corresponding to various binding energies of the alkali atoms (Na, K) to each other, whose leading term is taken to be their activation energy, $E_\alpha$. Following this formula, the entire material comprising a series of such layers has to maximise entropy, even for the mixed-alkali atom honeycomb layered oxides. As affirmed earlier, the highest entropy configuration leading to a minimised free energy and a stable crystalline structure is the one where both types of alkali atoms are arranged in a honeycomb fashion. The next favoured configuration is the one that allows for only one type of alkali atom to be in a honeycomb fashion, case in point being the configuration observed in $NaKNi_2TeO_6$ as displayed in Fig. 4a, b. Moreover, since the activation energy of K is lower than that of Na[1,34], it is apparent, by setting $F = 0$, that the Na layer can still minimise its free energy by disrupting its honeycomb configuration into e.g. triangular patterns shown in Fig. 4a. Thus, since the arrangement of the alkali atoms correlates with the adjacent metal slab atoms (Te, Ni), these slabs shear in order to accommodate the disruption, as observed in Fig. 2h.

A similar entropic and free energy argument can be made to account for sodium and potassium spontaneously intercalating between the oxide layers separately albeit adjacent to each other. This corresponds to new binding energy terms in the potential energy inversely proportional to the separation distance of atoms of the same type. The simplest term can be written as $U(d) \simeq -E_a - \alpha/d^n$ where α is a positive constant, $n$ a positive integer and $d$ the separation distance of the atoms (e.g. for the binding energy offsetting the Coulomb repulsion between two alkali atoms of the same type, $\alpha = q^2/4\pi\epsilon$ is the fine structure constant and $n = 1$, where $q$ is the charge of the adjacent alkali atom of the same type and $\epsilon$ is the permittivity across the distance, $d$). Thus, since a small separation distance, $d$ further lowers the free energy of the material, the lowest stable free energy configuration is where $d = 0.62$nm or $d = 0.55$nm corresponding to the case of the pure atom $K_2Ni_2TeO_6$ or $Na_2Ni_2TeO_6$ respectively, where the distances are correlated with the ionic radii of the respective alkali atoms. Moreover, this strengthens the above argument for the disruption of the honeycomb pattern for Na in $NaKNi_2TeO_6$, since this new binding energy term is smaller for Na atoms compared to the case for K atoms, by virtue of a smaller $d$ for Na atoms compared to K atoms. Finally, the next favoured

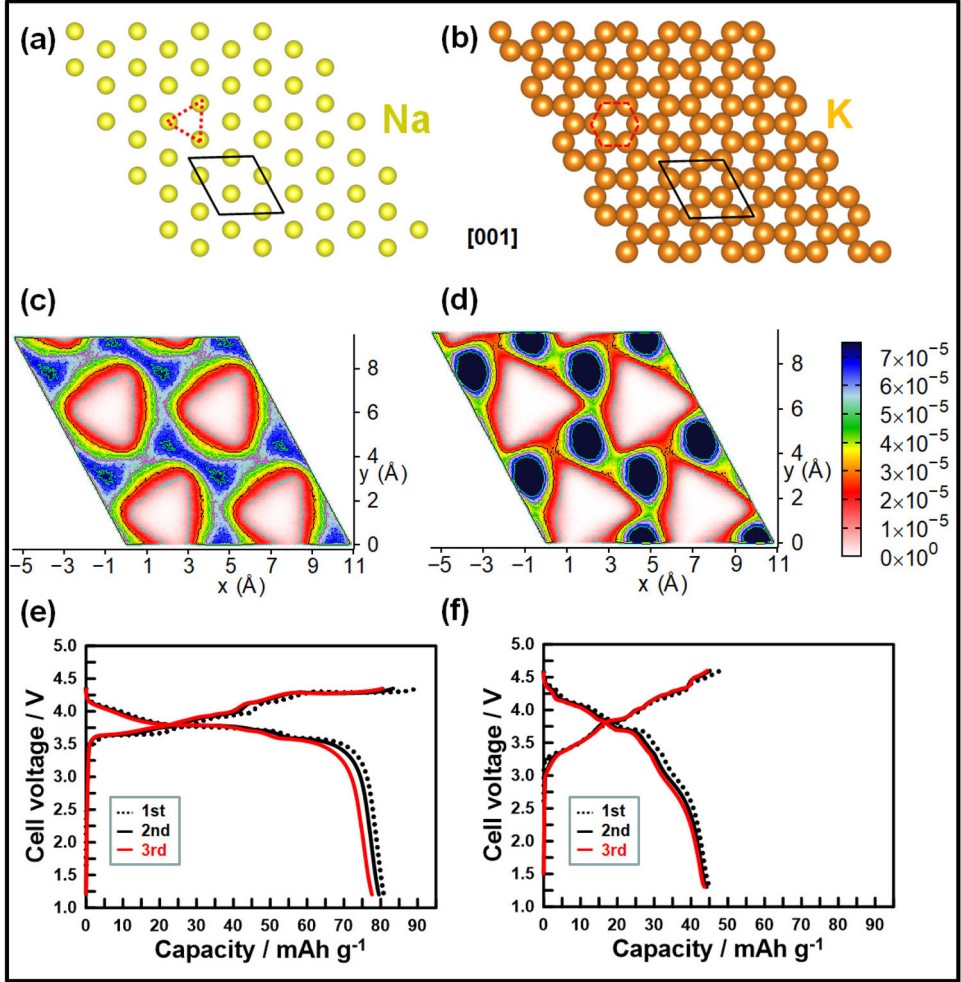

**Fig. 4 Alkali ion transport in NaKNi$_2$TeO$_6$. a** Arrangement of the Na atoms in triangular sites (as highlighted in red) as viewed along the [001] zone axis. The unit cell is highlighted in black lines. For clarity, the atoms denote the conformation of the sites and not the occupancy. **b** Honeycomb arrangement of K atoms in their respective sites as depicted along the [001] zone axis. **c** Molecular dynamics (MD) simulation showing the Na$^+$ ion probability density profile in the Na-layer (mapped onto 2 × 2 unit cells) using common colour bars (shown on the right). The population contours reflects that the preferred migration pathway amongst the interstitial cationic sites. **d** MD simulation for K$^+$ ion probability density profile in the K-layer. **e** Voltage profiles of the NaKNi$_2$TeO$_6$-based electrode tested using Na metal as counter electrode and Na$_{0.20}$Pyr$_{0.80}$FSI ionic liquid as electrolyte solution at 6.65 mA g$^{-1}$ and (**f**) Voltage profiles of the NaKNi$_2$TeO$_6$-based electrode tested using K metal as counter electrode and K$_{0.20}$Pyr$_{0.80}$FSI ionic liquid as electrolyte solution at 6.65 mA g$^{-1}$. Voltage–capacity profiles set at a lower cut-off voltage of 2.5 V have been furnished in the Supplementary Information (Supplementary Fig. 23).

configuration is given by, $d = (0.55 + 0.62)$ nm, corresponding to the mixed-alkali atom NaKNi$_2$TeO$_6$ reported herein, as shown in Fig. 2h.

Molecular dynamics (MD) simulations can avail insights into the microscopic alkali-ion transport of functional materials. This information is useful to gauge the feasibility of NaKNi$_2$TeO$_6$ as an energy storage material. MD simulations were performed based on the X-ray structure of NaKNi$_2$TeO$_6$, details of which are provided in the "Methods" section. The interaction between pairs of atoms was chosen from the previous report[16,17,35], as listed in Supplementary Table 5, and a few of them (indicated by footnote in Supplementary Table 5) are suitably modified to reproduce the desired bond lengths and coordination numbers. The employed interatomic potential model reproduces the structure satisfactorily (cell parameters variation (~3%), radial distribution function, as shown in Supplementary Table 6 and Supplementary Fig. 21 in the Supplementary Information). Computing the mean squared displacement at 600 K, we find that Na$^+$ ion has a higher diffusion coefficient ($D$) than K$^+$ ion based on the equation $\lim_{t \to \infty} <[\triangle r(t)]^2> = 4Dt$, (where $t$ is the time variable and $\triangle r(t)$ is

the displacement of the cation) inside the conduction layer as is evident in Supplementary Fig. 22. The Na$^+$ and K$^+$ ion population profile displayed in Fig. 4c, d also show the similar behaviour within the structure of NaKNi$_2$TeO$_6$. A few high-density areas are identified in the population profile, indicating favourable sites of Na$^+$ or K$^+$ ions. Particularly, the high-density areas are well connected for the Na-layer, whereas a modest connectivity amongst the high-density sites in K-layer is observed, resulting in higher diffusion of Na$^+$ ion compared to K$^+$ ion. It is worth to mention that this behaviour is different than the parent Na- or K-systems i.e. $A_2$Ni$_2$TeO$_6$, where $A$ = Na, K. This can be traced to the vastly different NiO$_6$ and TeO$_6$ octahedral stacking sequences from the parent Na or K-systems, which leads to a differing local environment[1,34]. Recall, we observed in STEM (Fig. 2g) that, the layers where K atoms occupy the interlayer space, Te/Ni slabs are not shifted with respect to each other whereas, for layers where Na atoms reside, shifts of the Te/Ni slabs are observed. This behaviour is maintained in the simulation results despite high temperatures where the alkali ions are dynamic. Supplementary Video 1 indeed shows the dynamic

behaviour of $Na^+$ and $K^+$ ions, when simulated at 600 K. Further investigation of the nature of $Na^+$ and $K^+$ ion transport and its mechanism is beyond the scope of this work.

Presumably, $NaKNi_2TeO_6$ like other honeycomb layered oxides, holds potential in many fields. Nonetheless, the primary focus of this study is to ascertain its feasibility as positive electrode active material for alkali-metal batteries. Thus, electrochemical energy storage tests were carried out to verify its ability to transport and store mixed-alkali ions. For such reasons, Na and K half-cells were assembled, as further explicated in the "Methods" section and Supplementary Information (Supplementary Fig. 23). Even though tellurium is not a constituent element of choice for energy storage systems entailing Earth-abundant elements, the insights obtained herein are not exclusive to the mixed-alkali tellurate systems. Indeed, the present study should be considered as a fundamental scientific research work rather than an applied one. Moreover, we do not rule out the use of tellurates in niche applications where functionality may be prioritised over cost. Figure 4e shows the voltage–capacity plots of $NaKNi_2TeO_6$ in Na half-cells. The theoretical capacity for a full $Na^+$ ion extraction from $NaKNi_2TeO_6$ is approximately 67 mAh $g^{-1}$. However, a reversible capacity of ca. 80 mAh $g^{-1}$ was attained upon subsequent cycling, suggesting the occurrence of $K^+$ ion extraction. In the case of the K half-cells (Fig. 4f), an initial capacity of 45 mAh $g^{-1}$ was realised and maintained upon successive cycling. This capacity presumably arises from predominant $K^+$ ion extraction given that K metal was used.

Post-mortem imaging of $NaKNi_2TeO_6$ electrodes subsequently cycled in Na- and K-half-cells were performed using high-resolution STEM, in order to ascertain the nature of the intercalation and de-intercalation process of the alkali ions. Figure 5a shows the ex situ HAADF-STEM images of a $NaKNi_2TeO_6$ electrode upon subsequent cycling (i.e., fully discharged sample at the third cycle) in Na-half-cells taken at the [100] zone axis and the corresponding ABF-STEM images are shown in Fig. 5b. SAED patterns taken along the [100] axis are shown in Fig. 5c. Subsequent cycling of $NaKNi_2TeO_6$ in Na half-cells leads to the replacement of $K^+$ with $Na^+$ ions to yield a Na-rich phase composition ($Na_2Ni_2TeO_6$), as affirmed by the equidistant interlayer spacings (0.55 nm) of Na atoms along the [001] axis as quantitatively illustrated by the intensity line profiles (Fig. 5d) for the highlighted area in the HAADF-STEM images (shown in Fig. 5a). Streaks are evinced in the SAED patterns taken at the [100] axis (Fig. 5c), indicating modulation in the arrangement of Na atoms along the *ab* plane as is exemplified in $Na_2Ni_2TeO_6$[36].

Figure 5e, f show, respectively, the ex situ HAADF- and ABF-STEM micrographs of a $NaKNi_2TeO_6$ electrode upon cycling in K-half-cells taken at [100] zone axis. Figure 5g shows the corresponding SAED patterns. Alternating interlayer spacings (0.55 and 0.62 nm) of Na and K atoms along the [001] axis are observed, indicating that the mixed-alkali layered framework is retained owing to reversible extraction and insertion of $K^+$ ion alone. Voltage–capacity plots of $NaKNi_2TeO_6$ upon subsequent cycling in K half-cells reveal a reversible initial capacity of approximately 50 mAh $g^{-1}$ (Fig. 4f). The theoretical capacity for a full $K^+$ extraction from $NaKNi_2TeO_6$ is approximately 67 mAh $g^{-1}$, which indicates that the capacity arises predominantly from $K^+$ ion extraction. Given that K metal was used as anode (counter electrode), reversible extraction and reinsertion of $K^+$ ions can be envisaged, as is validated experimentally judging from the intensity line profile (Fig. 5h) that quantitatively show alternating interlayer distances occupied by Na and K atoms. Further, SAED patterns (Fig. 5g) show streaks indicative of the aperiodic stacking nature along the *c*-axis.

These electrochemical measurements indicate that $NaKNi_2TeO_6$ mixed-alkali honeycomb layered oxide is amenable to electrochemical binary alkali-ion transport and storage, pointing towards the possibility of developing a viable mixed $Na^+$- and $K^+$-ion electrochemical cell that relies on electrolytes and electrode materials that can accommodate both $Na^+$ and $K^+$ binary-cation transport. Given that cells utilising both cation and anion as charge carriers (dual-ion batteries (DIBs)) have already shown remarkable metrics in terms of energy density, power density and cycling life[37–39], the present battery chemistry exploiting binary alkali metal cations could be a promising successor to DIB technology[38,40–42]. Indeed, taking into account the abundance of Na and K, a cell with suitable specific energy and cyclability can be designed. Moreover, it offers the possibility of utilising a NaK liquid metal alloy (albeit at its nascent stage of development) as anode material which can be effective in accommodating the cations, thwarting the formation of dendrites that have long plagued the direct utilisation of alkali metal anodes in secondary batteries (as illustrated in Fig. 6a)[41,42]. Voltage–capacity plots of $NaKNi_2TeO_6$ when initially cycled in a NaK half-cell, using a mixed electrolyte based on a $Na_{0.10}K_{0.10}Pyr_{0.80}FSI$ ionic liquid, displays a reversible capacity of about 80 mAh $g^{-1}$ at an average voltage of approximately 3.8 V (Fig. 6b). The theoretical capacity of $NaKNi_2TeO_6$ is 134 mAh $g^{-1}$, assuming a full extraction of $Na^+$ and $K^+$ ions in a NaK cell. Although about 60% of theoretical capacity is attained, the performance is promising considering no electrode optimisation (carbon-coating, nanosizing, etc.) has been undertaken. The corresponding voltage–capacity plots at various specific currents are shown in Fig. 6c, indicating $NaKNi_2TeO_6$ sustains decent rate capabilities. Moreover, reversible electrochemical behaviour is observed for 20 cycles at a specific current of 13.4 mA $g^{-1}$, as shown in Fig. 6d, and with good Coulombic efficiency (Supplementary Fig. 27). To fully tap the potential of $NaKNi_2TeO_6$, further electrode optimisation strategies are warranted, which is a subject of future work. Ionic liquids were utilised owing to their higher stability at high-voltage regimes, in comparison to ether- and ester-based solvent electrolytes[43,44,45]. Figure 6e shows voltage–capacity comparison plots of $NaKNi_2TeO_6$ along with cathode materials reported for NaK battery system. $NaKNi_2TeO_6$ is the first material to have both Na and K atoms initially stabilised in its layered framework and offers a high discharge voltage along with relatively high capacity. The high capacity of $NaKNi_2TeO_6$ could be associated with the redox process of Ni. Indeed, X-ray photoelectron spectroscopy measurements (shown in Supplementary Fig. 28) indicate the participation of Ni to the charge compensation process, whereas Te is dormant, as has been noted in related honeycomb layered oxides such as $K_2Ni_2TeO_6$ and $Na_4NiTeO_6$[2,4]. With a judicious choice of constituent elements, we speculate that mixed-alkali compositions could exhibit even higher voltage and capacity.

Since both $Na^+$ and $K^+$ ions reinsertion can be envisaged during cycling of $NaKNi_2TeO_6$ in NaK cells, further structural insights were attained from high-resolution STEM ex situ measurements. Figure 7a, b show the ex situ HAADF-STEM images along the [100] and [1$\bar{1}$0] zone axes, respectively, for a $NaKNi_2TeO_6$ crystallite taken after subsequent cycling (i.e., fully discharged sample at the third cycle) in NaK cell. Equidistant interlayer spacings are apparent, suggesting that one type of alkali-ion is reversibly reinserted into $NaKNi_2TeO_6$ upon successive cycling. Intensity line profiles reveal equidistant interlayer spacings (0.55 nm) corresponding to Na atoms in the lattice (Fig. 7c and Supplementary Fig. 29). Further, SAED patterns taken along the [100] and [1$\bar{1}$0] zone axes (Fig. 7d, e) evidence streaks reminiscent of the modulation in the Na atoms arrangement in Na-rich phase ($Na_2Ni_2TeO_6$)[36]. Whilst only Na-rich phases are observed in TEM measurements, ex situ XRD measurements of discharged electrodes (Supplementary Fig. 30) reveal K-rich and mixed-alkali phases with Na-rich phase being

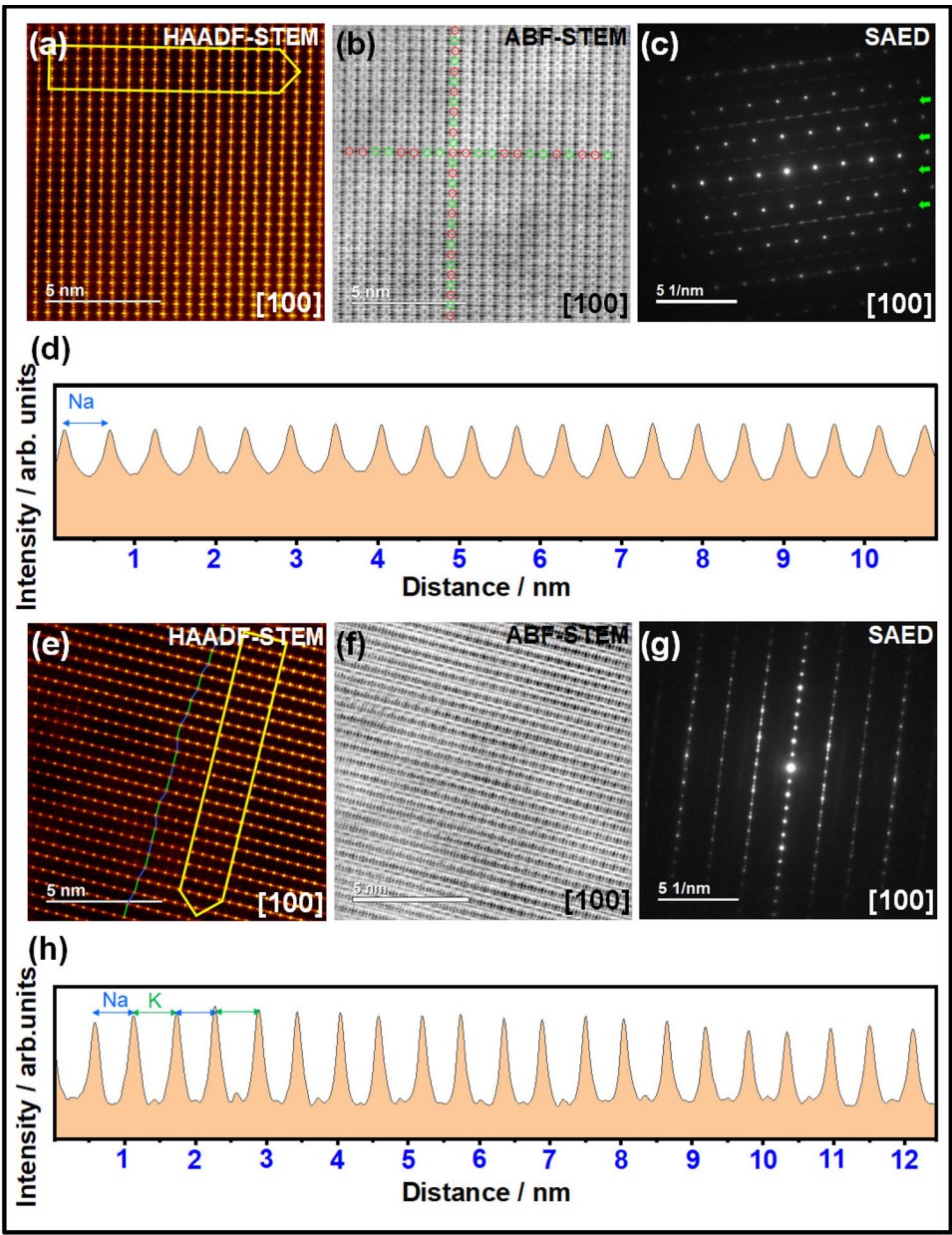

**Fig. 5 Structural changes during alkali-ion (de)insertion in NaKNi$_2$TeO$_6$. a** HAADF-STEM ex situ images of NaKNi$_2$TeO$_6$ taken along the [100] axis upon cycling in a Na half-cell (i.e., fully discharged at the third cycle). **b** Corresponding ABF-STEM images and (**c**) SAED patterns showing streaks (shown in green arrows) indicative of the modulation in the Na arrangement along the *ab* plane as is exemplified in Na$_2$Ni$_2$TeO$_6$. **d** Intensity line profiles as highlighted in (**a**) showing equidistant interlayer spacings (0.55 nm) of Na atoms along the [001] axis. **e** HAADF-STEM ex situ images of NaKNi$_2$TeO$_6$ taken along the [100] axis upon cycling in a K half-cell (i.e., fully discharged at the third cycle). **f** Corresponding ABF-STEM images and (**g**) SAED patterns showing streaks indicative of the aperiodic stacking nature along the *c*-axis. **h** Intensity line profiles as highlighted in (**e**) showing alternating interlayer spacings (0.55 nm and 0.62 nm) of Na and K atoms, respectively, along the [001] axis. For clarity, the horizontal axes in (**d**) and (**h**) show the number of layers. Intensity line profiles that quantitatively illustrate the interlayer spacing values shown in (**d**) and (**h**) are provided as Supplementary Information (Supplementary Figs. 24, 25).

predominant. These results reveal that there is a propensity of NaKNi$_2$TeO$_6$ to predominantly reinsert Na ions when cycled in NaK cells. Presumably, this could be rooted in the stability of Na$_2$Ni$_2$TeO$_6$ over other phases, as affirmed by theoretical computations (Supplementary Figs. 31–33) that show the stability of Na-rich Na$_2$Ni$_2$TeO$_6$ over other phases such as NaKNi$_2$TeO$_6$ and K$_2$Ni$_2$TeO$_6$. A close inspection of Fig. 7a further reveals a variation in the Te/Ni stacking of the Na-rich phases, with domains where the slabs do not shift and domains where they do (highlighted by green arrows in Fig. 7a) are observed. A rationale for the shear transformations has been appended as Supplementary Note 1.

It is imperative to mention that the concept of mixed-alkali battery materials can be beneficially applied to tailor the ability to transport and store alkali metal ions. For instance, the partial substitution of Li atoms in layered transition metal cathode oxides with Na, K, Rb or Cs is a well-investigated route to enhance their structural stability and increase Li-ion diffusion[46–54]. However, the theoretical capacity of materials with equimolar amounts (i.e., 50% atomic fraction) of different alkali metal atoms, such as NaKNi$_2$TeO$_6$, is drastically attenuated when large amounts of Na atoms are replaced with K atoms in a cathode for a Na-battery or vice versa. This is generally ascribed to the fact

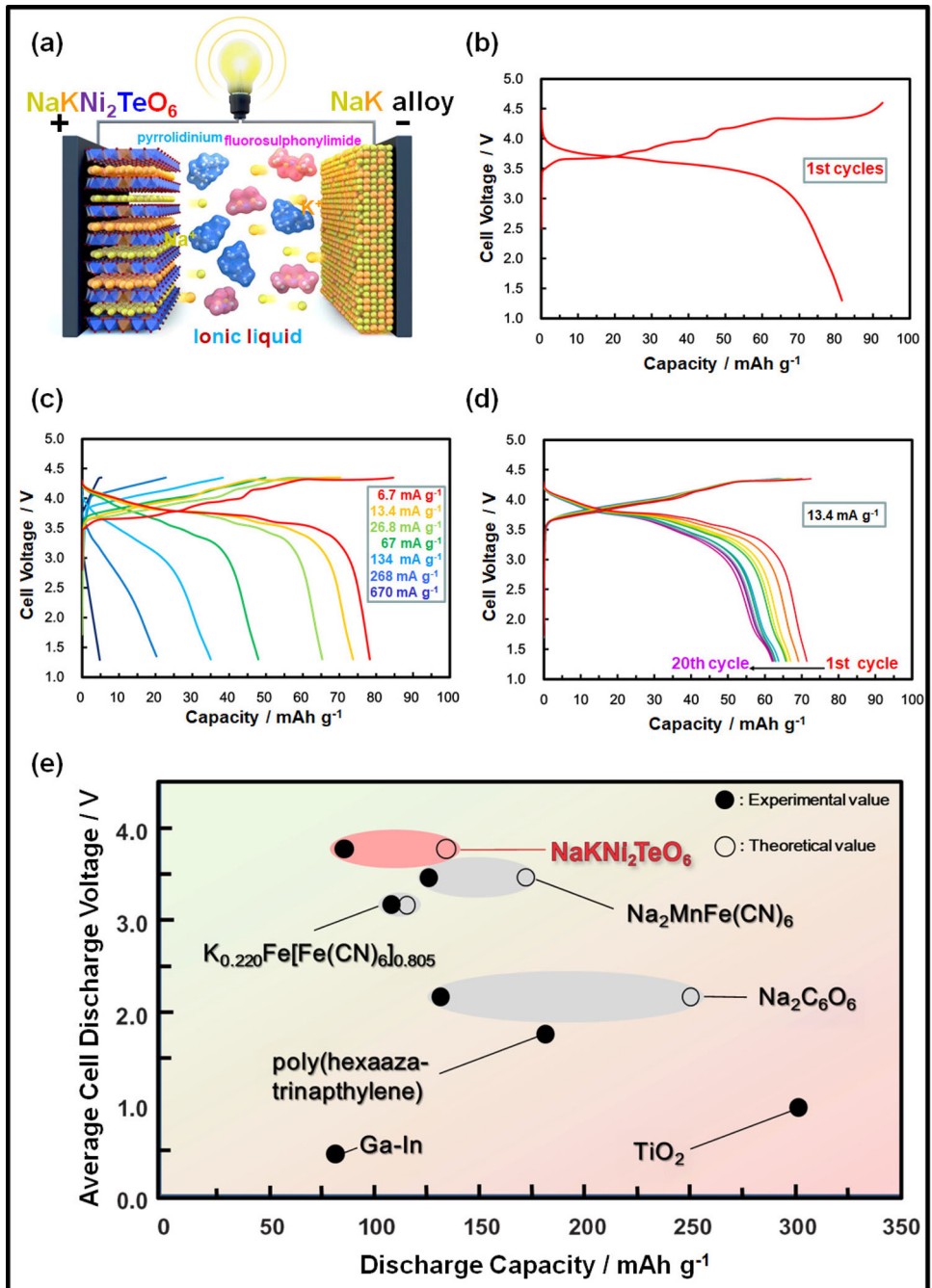

**Fig. 6 Electrochemical energy storage performances of NaKNi$_2$TeO$_6$ as a cathode active material for NaK cells. a** A schematic illustration of a dual (Na- and K-ion) battery using NaKNi$_2$TeO$_6$ as cathode and dendrite-free NaK liquid alloy as anode. **b** Voltage–capacity profiles of NaKNi$_2$TeO$_6$ dual-cation cathode material in a cell using NaK as anode during initial cycling at a specific current of 6.7 mA g$^{-1}$. The voltage range was set at 4.6–1.3 V. The electrolyte used was a pyrrolidinium-based dual-metal-cation ionic liquid, comprising equimolar amounts of Na and K (i.e., Na$_{0.10}$K$_{0.10}$Pyr$_{0.80}$FSI). Details regarding the cell assembly are provided in the "Methods" section. **c** Voltage–capacity profiles at various specific currents under a voltage range of 4.35–1.3 V. **d** Cyclability performance at a specific current of 13.4 mA g$^{-1}$ rate upon subsequent cycling (20 cycles). **e** Voltage–capacity plots of cathode materials reported so far for NaK batteries. Comparison data of NaKNi$_2$TeO$_6$ with reported cathode materials for NaK batteries are furnished as Supplementary Table 7. For clarity, the methodology to calculate the average discharge voltage is reported in the Supplementary Information section (Supplementary Fig. 26).

that the number of cations participating in extraction and reinsertion would be diminished. As a way to enhance performance, the utilisation of an alloy such as NaK in the case of NaKNi$_2$TeO$_6$ would facilitate the participation of both cation species thus yield high theoretical capacity. A schematic showing the conceptual design of such a battery is shown in Fig. 6a. In addition, the liquid nature of NaK does not allow the formation of dendrite on the

anode thus rendering the design 'a dendrite-free' metal anode cell[38,40]. Therefore, this concept showcases the potential for NaKNi$_2$TeO$_6$ and related mixed-alkali layered oxide materials as functional materials.

Motivated by the projected functionalities of such materials, attempts were made to synthesise various compositions that form structures akin to those of NaKNi$_2$TeO$_6$. Partial substitution of Ni

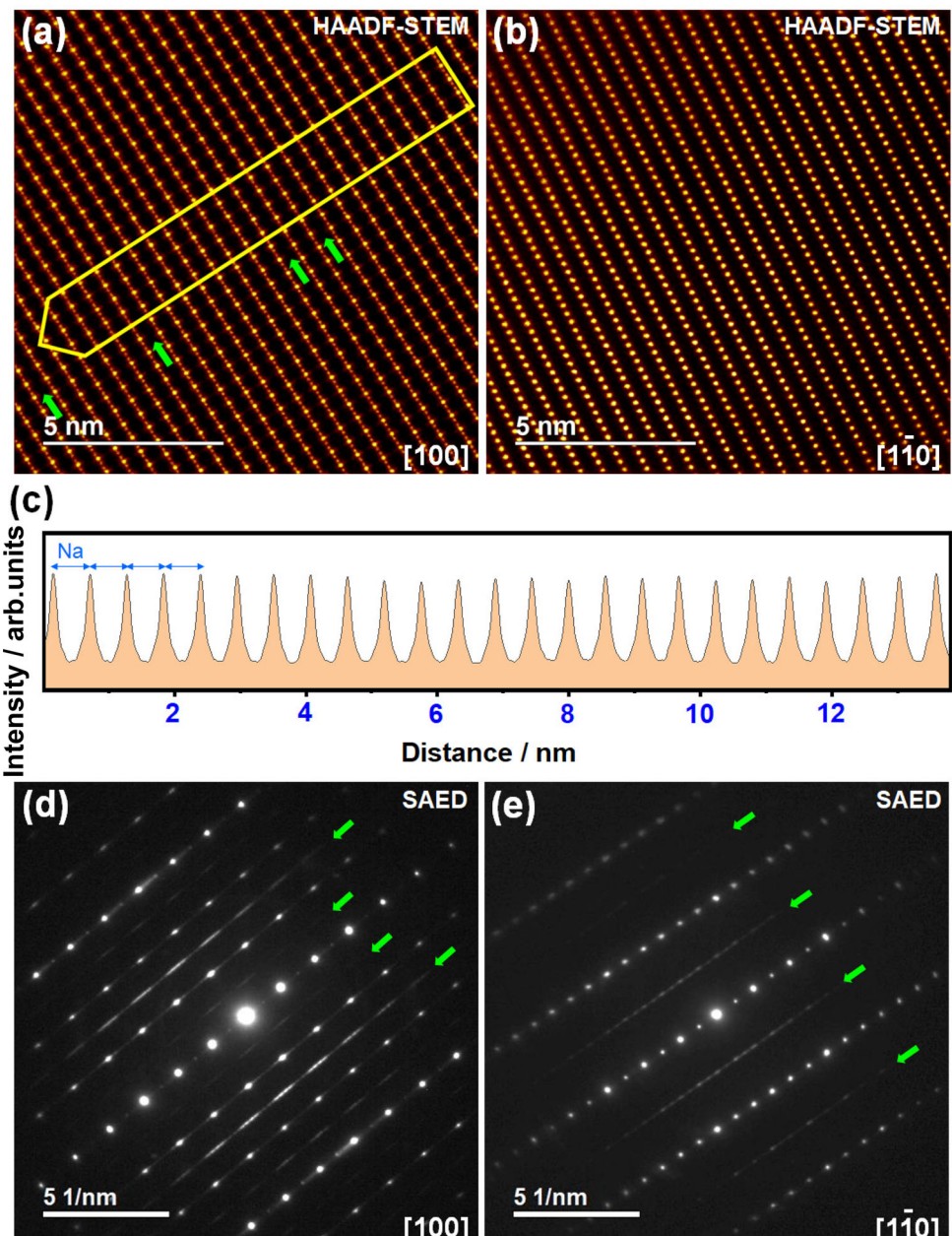

**Fig. 7 Structural changes during cycling of NaKNi₂TeO₆ in NaK batteries.** HAADF-STEM ex situ images of $NaKNi_2TeO_6$ upon complete discharging for three cycles in NaK cells taken along (**a**) [100] axis and (**b**) [1$\bar{1}$0] zone axis. Green arrows show Na atom layers where the adjacent Te/Ni slabs shift. A rationale for the slab shear transformations has been provided as Supplementary Note 1. **c** Intensity line profiles as highlighted in (**a**) showing equidistant interlayer spacings (0.55 nm) along the [001] axis revealing $NaKNi_2TeO_6$ to preferentially insert Na atoms into the lattice. SAED patterns taken along (**d**) [100] axis and (**e**) [1$\bar{1}$0] axis showing streaks (underpinned in green arrows) indicative of the modulation in the Na arrangement as is exemplified in $Na_2Ni_2TeO_6$[37].

with Co, lead to the successful synthesis of mixed-alkali layered oxides adopting the compositions $Na_{2-x}K_xNi_{2-y}Co_yTeO_6$ ($y$ = 0.25, 0.5, 0.75, 0.1) (as shown in Supplementary Figs. 34, 35). Material characterisation using XRD data confirm the formation of intermediate phases regardless of the extent of Co-doping, highlighting the possibility to create structures similar to that of $NaKNi_2TeO_6$ in other compositions such as $NaKM_2TeO_6$ ($M$ = Cu, Zn, Co) as shown in Supplementary Fig. 36. In addition, this work has also accentuated the further exploration of other mixed-alkali honeycomb layered oxides that entail alkali atoms such as Li (as shown in Supplementary Fig. 37).

In conclusion, the successful design of mixed-alkali honeycomb layered oxides, for instance $NaKNi_2TeO_6$, not only offers a conduit to engineering new functional materials but also promises to expand the compositional space of known honeycomb layered oxides. The results of this study reaffirm the correlation between the ionic radii of the alkali atoms and the interlayer distance even for the mixed-alkali system, which can be exploited to configure the intricate interlayer structure of the mixed-alkali honeycomb layered oxides, by the same token as $(Li/Ag)CoO_2$[55]. Detailed local atomic information provided through a series of scanning transmission electron microscopy reveal characteristics of a

unique aperiodic stacking structure, suggesting structural versatility that could unlock the potential of this material for electromagnetic, quantum and electrochemical functionalities[1,32,56]. Further, we expound on the feasibility of $NaKNi_2TeO_6$ for battery applications that utilise mixed cation transport. The mixed triangular and honeycomb atomics conformations may have profound impact on the electrodynamics of the alkali ions. An attempt to rationalise this view has been made by theoretical computations. We hope that this work will serve as a cornerstone for further augmentation of mixed-alkali layered oxides in various realms of science and technology.

## Methods

**Synthesis of materials**. Honeycomb layered oxide samples with nominal compositions $Na_{2-x}K_xNi_{2-y}Co_yTeO_6$ ($0 \leq x \leq 2$; $y = 0$, 0.25, 0.5, 0.75 and 1) were prepared through high-temperature solid-state reaction routes. Stoichiometric amounts of CoO (Sigma Aldrich/High Purity Chemicals), NiO (Kojundo Chemical Laboratory (Japan), purity of 99%), $TeO_2$ (Aldrich, purity of ≥99.0%), $Na_2CO_3$ (Chameleon Reagent) and $K_2CO_3$ (Rare Metallic (Japan), purity of 99.9%) were thoroughly pulverised (using an agate mortar and a pestle) to attain the precursors for $Na_{2-x}K_xNi_{2-y}Co_yTeO_6$. Powders of these precursors were thereafter pelletised and annealed in gold crucibles at temperature ranges of 800–840 °C in air for 24 h. To avert any exposure to moisture, the as-prepared materials were transferred to an argon-purged glove box. A phase-pure $NaKNi_2TeO_6$ could be prepared via a direct solid-state reaction of $Na_2Ni_2TeO_6$ and $K_2Ni_2TeO_6$ in air.

Related mixed-alkali honeycomb layered compositions embodied by $NaKM_2TeO_6$ (where $M$ = Cu, Co, Zn) were synthesised using $K_2CO_3$ (Rare Metallic (Japan), purity of 99.9%), $Na_2CO_3$ (Chameleon Reagent), $Co_2O_3$ (Kojundo Chemical Laboratory (Japan), CuO (Kojundo Chemical Laboratory (Japan), 99%), ≥99.0%) and ZnO (Wako Chemicals (Japan), 95%) in the temperature ranges of 800 to 840 °C in air or Ar for 24 h. There is a propensity to improve the purity of the samples via the use of excess amount of alkali metal carbonates. The XRD patterns for the as-prepared $NaKM_2TeO_6$ (where $M$ = Cu, Co, Zn) are furnished as Supplementary Information (Supplementary Fig. 36).

**X-ray diffraction (XRD) measurements and analysis**. Conventional XRD (CXRD) measurements were conducted using a Bruker D2 PHASER diffractometer employing a Cu–Kα radiation (viz., $\lambda = 1.54056$ Å). CXRD measurements were performed in Bragg-Brentano geometry under a $2\theta$ range of 5° ~ 90° with a step size of 0.01°. Analysis of the CXRD data was carried out by fitting protocols implemented in the JANA 2006 program[57].

Synchrotron XRD (SXRD) measurements were performed to acquire high-resolution data for analyses of the crystal structure of $NaKNi_2TeO_6$. SXRD experiments were performed at BL8S2 of Aichi SR centre. Time-of-flight (TOF) powder neutron diffraction (ND) data of $NaKNi_2TeO_6$ samples were collected at room temperature using the SPICA diffractometers installed at the Material and Life science Facility (MLF) in the Japan Proton Accelerator Research Complex (J-PARC). The powder samples were sealed in cylindrical vanadium cells of the following dimensions: 40 mm in height and 5.8 mm in diameter. Neutron diffraction data taken using the backscattering bank were evaluated and refined using the FULLPROF suite, JANA2006, and Z-Rietveld softwares[57,58]. VESTA was used to display the refined crystal structures[59]. Full crystallographic data was curated in the joint CCDC/Fiz Karlsruhe Crystal Structure Database under CSD-2070815. Moreover, modelling and quantitative analyses of the stacking faults innate in $NaKNi_2TeO_6$ were done using the FAULTS program[31].

XRD ex situ measurements of pristine and discharged electrodes in Na, K and NaK half-cells were collected in Bragg–Brentano geometry using a Cu Kα monochromator. Electrochemical measurements were stopped upon fully discharging the $NaKNi_2TeO_6$ electrodes at 1.3 V at the third cycle and cells dismantled in an argon-filled glove box (water and oxygen concentration maintained at below 1 ppm). Prior to performing XRD measurements, the discharged electrodes were washed using 25 mm of super-dehydrated acetonitrile (water content of <10 ppm) for five times and subsequently dried inside an argon-purged glove box. XRD measurements were thereafter performed without exposing the samples to moisture using a sample holder (designed by Bruker AXS Limited).

**Morphological and chemical characterisations**. Chemical compositions and quantifications were determined by inductively coupled plasma absorption electron spectroscopy (ICP–AES) on a Shimadzu ICPS-8100 instrument. Repeated measurements were done on several different spots of the samples, with no deviation observed in the calculated values (Supplementary Table 2).

As for high-resolution transmission electron microscopy (TEM) observation, the pristine powder particles were embedded in epoxy glue under an Ar-atmosphere and then thinned by an Ar-ion-milling method using a GATAN PIPS (Model 691) precision ion-milling machine. Specimens were transferred into the apparatuses with minimal exposure to air. As for (dis)charged electrodes, powder particles were scratched from the current collector metal foil after drying.

High-resolution scanning TEM (STEM) imaging and electron diffraction patterns were obtained using a JEOL JEM-ARM200F with a CEOS CESCOR STEM Cs corrector (spherical aberration corrector) operated at an acceleration voltage of 200 kV. Measurements were performed along three-zone axes with identical particles in [100] and [$1\bar{1}0$] zone axes, and a different particle in the [001] zone axis. Owing to the vulnerability of the specimens to electron irradiation, the nominal STEM probe current of electron-beam dose was maintained at a relatively low value of 23 pA under short-exposure times. The nominal probe convergent semi-angle was about 20 mrad. High-angle annular dark-field (HAADF) and annular bright-field (ABF) STEM images were obtained simultaneously at nominal collection angles of 90–370 mrad and 11–23 mrad, respectively. To avoid the image distortion due to drift of the specimen during a scan, an "integration of quickly acquired images" method was conducted for the observation of atomic structures[60]. About 20 STEM images were recorded sequentially with an acquisition time of about 0.5 s per image, and then the images were aligned and superimposed to one image. This method is also effective to reduce damage caused by electron beam since the local heat accumulation caused by the beam dwelling in a constricted area is reduced. Electron diffraction experiments were conducted on many crystallites, and reproducible results were observed. The images showing detailed atomic structures were composed by averaging about 20 small images extracted from different areas of the larger images in order to increase signal-to-noise ratio. This allowed an explicit localisation of transition metal atomic columns in the obtained STEM maps. STEM-EDX (energy dispersive X-ray spectroscopy) spectrum images were obtained with two JEOL JED 2300 T SDD-type detectors with 100 mm² detecting area whose total detection solid angle was 1.6 sr. Elemental maps were extracted using Thermo Fisher Scientific Noran (NSS) X-ray analyser.

**Thermal stability measurements**. Thermogravimetric and differential thermal analysis (TG-DTA) was performed using a Bruker AXS 2020SA TG-DTA instrument in the temperature ranges of 25 to 900 °C. The measurements were performed at a ramp rate of 5 °C min$^{-1}$ under argon using a platinum crucible. Thermal stability measurements reveal $NaKNi_2TeO_6$ mixed-alkali honeycomb layered oxide to be stable to up to 800 °C (Supplementary Fig. 9), as further confirmed by XRD measurements (see Supplementary Fig. 10).

**Electrochemical measurements**. Coin cell assembly protocols were performed in an Ar-purged glove box (MIWA, MDB-1KP-0 type) with $H_2O$ and $O_2$ contents less than 1 ppm. For electrode fabrication, pristine $NaKNi_2TeO_6$ was mixed with carbon black and polyvinylidene fluoride (PVdF) to attain a final weight ratio of active material: carbon: binder in the cathode of 70: 15: 15. The mixture was suspended in $N$-methyl-2-pyrrolidinone (NMP) to obtain a viscous slurry, which was then cast on tungsten foils with a typical mass loading of ~5 mg cm$^{-2}$. Note that tungsten foil was preferred to aluminium foil as current collector, owing to its high oxidative stability. Electrodes with a geometric area of 1 cm² were punched and dried at 120 °C in a vacuum oven. The average thickness of the electrodes was around 50 μm. Electrochemical properties of the materials were evaluated in 2032-type coin cells using $NaKNi_2TeO_6$-based positive electrode (i.e., the working electrode) separated from the K, Na or NaK metal negative electrodes (i.e., the counter/reference electrodes) by glass fibre discs soaked in electrolyte. The total volume of electrolyte used per coin cell was 80 microlitres. Sodium-potassium (NaK) liquid alloy was prepared at room temperature in an argon-purged glove box. Sodium and potassium were mixed physically in a weight percentage of 54.1 and 40.9, respectively, in a glass vial. The shiny sodium-potassium alloy was formed spontaneously upon mixing sodium (purity of 99 %, Kishida Chemicals) and potassium (purity of 99.95%, Kishida Chemicals) lumps. Sodium-potassium alloy was immersed in super-dehydrated hexane (water content of <10 ppm) in order to avert the formation of oxides at the surface. A glass syringe was used to take a portion of the liquid alloy for assembly of the cells. Sodium-potassium alloy (which has a high surface tension) was immobilised in the coin cells using aluminium meshes (100-mesh size and purity of 99% (Nilaco Corporation)). For clarity, 2032-type coin cells are of the following dimensions: diameter of 20 mm and a thickness (height) of 3.2 mm. The electrolyte used was a 1.0 mol dm$^{-3}$ potassium bis(fluorosulphonyl)imide (KFSI) in $1$-methyl-$1$-propylpyrrolidinium bis(fluorosulphonyl)imide ($Pyr_{13}FSI$) (Kanto Chemicals (Japan), 99.9%, <20 ppm $H_2O$) ionic liquid for K half-cells[2,3,8]. A 1 mol dm$^{-3}$ sodium bis(fluorosulphonyl) imide (NaFSI) in $Pyr_{13}FSI$ ionic liquid was used for Na half-cells[43,44], whereas a 0.5 mol dm$^{-3}$ equimolar mixture of NaFSI and KFSI in $Pyr_{13}FSI$ ionic liquid was used in assembling NaK cells[39,45]. Protocols relating to the preparation of these electrolytes have been detailed elsewhere[8,43,44]. Molar ratio of the ionic liquids was calculated based on their molecular weight and density at 298 K. The densities for 0.5 mol dm$^{-3}$ NaFSI + 0.5 mol dm$^{-3}$ KFSI/$Pyr_{13}$FSI, 1.0 mol dm$^{-3}$ NaFSI/$Pyr_{13}$FSI, 1.0 mol dm$^{-3}$ KFSI/$Pyr_{13}$FSI, and 0.5 mol dm$^{-3}$ KTFSI/$Pyr_{13}$TFSI were 1.4323, 1,4119, 1.4227, and 1.4283 g cm$^{-3}$, respectively. The concentration (expressed in molar ratio) of the electrolytes are respectively as follows: $Na_{0.10}K_{0.10}Pyr_{0.80}FSI$, $Na_{0.20}Pyr_{0.80}FSI$, $K_{0.20}Pyr_{0.80}FSI$ and $K_{0.13}Pyr_{0.87}TFSI$. For clarity to readers, pyrrolidinium-based ionic liquid is abbreviated in literature as Pyr(r)$_{13}$TFSA (Pyr (r)$_{13}$TFSI or Pyr(r)$_{13}$Tf$_2$N)).

The physicochemical properties of the mixed ionic liquid (0.5 mol dm$^{-3}$ NaFSI + 0.5 mol dm$^{-3}$ KFSI in $Pyr_{13}$FSI) used, in the present study, as an electrolyte for the NaK half-cell are shown in Supplementary Table 8. Galvanostatic cycling tests were

carried out applying specific currents ranging from 6.7 mA g$^{-1}$ to 670 mA g$^{-1}$. Unless otherwise stated, the cut-off voltage was set at 1.2 V to 4.35 V for the Na half-cells or 1.3 V to 4.6 V as for the K half-cells. A cut-off voltage of 1.3 V to 4.35 V was set for the sodium-potassium full-cells. All the electrochemical measurements were performed at room-temperature (25 ± 1°C) with temperature maintained using a temperature-controlled oven (ESPEC).

**Theoretical computations**. Molecular dynamics (MD) simulations at constant pressure and temperature ($N, P, T$) were carried out using the Vashishta–Rahman type interatomic pair potential that has been used remarkably well for a variety of system including related honeycomb layered oxides such as Na$_2$$M_2$TeO$_6$ ($M$ = Mg, Zn, Co, Ni)[16,17,35],

$$U\left(r_{ij}\right) = \frac{q_i q_j}{4\pi\varepsilon_0 r_{ij}} + \frac{A_{ij}(\sigma_i + \sigma_j)^{n_{ij}}}{r_{ij}{}^{n_{ij}}} - \frac{P_{ij}}{r_{ij}{}^4} - \frac{C_{ij}}{r_{ij}{}^6} \quad (1)$$

where $q_i$ is the charge and $\sigma_i$ is the ionic radius of the $i$th ion. $r_{ij}$ is the interatomic distance. The parameters, $A_{ij}$, $P_{ij}$, and $C_{ij}$, are the short-range interaction parameters, between ion pairs $i$ and $j$. Simulations were done using the software package LAMMPS[61]. The pressure and temperature in the system were controlled using Nose-Hoover type thermostatting and barostatting techniques. The interaction between pairs of atoms was chosen from the previous report[16,17,35], as listed in Supplementary Table 5, and a few of them (indicated by footnote in Supplementary Table 5) are suitably modified to reproduce the desired bond lengths and coordination numbers. The employed interatomic potential model reproduces the structure satisfactorily (cell parameters variation (~3%), radial distribution function, as shown in Supplementary Table 6 and Supplementary Fig. 21 in the Supplementary Information section). We chose a hexagonal simulation box containing 2156 atoms, as found in the XRD pattern obtained in the present study, and applied periodic boundary conditions in all three Cartesian directions. Newton's equations of motion were integrated using the velocity form of Verlet's algorithm with a time step of 2 fs. The typical duration of an MD run was 6 ns. A particle-particle-particle-mesh $k$-space solver was used to compute long-range van der Waals and Coulomb interactions beyond a cut-off distance of 11 Å at each time step. Furthermore, we performed a simulation by freezing Te at their crystallographic positions to stop the layers sliding. A constant volume and temperature (600 K) MD simulation was also performed from the final structure of NPT-MD simulation for the occupancy profile calculation, as shown in Fig. 4c, d. The coordinates of Na$^+$ and K$^+$ ions were folded back into a unit cell on two-dimensional fine grid spanning on the conduction plane (ab plane) and averaged over almost 30000 configurations over a 6 ns long NVT–MD simulation at 600 K. The occupancy patterns were then replicated over 2 × 2-unit cells for ease of visualisation and its connectivity. The total occupancy in each layer was normalised to unity.

Structural stability of the mixed-alkali honeycomb structure is studied by calculating the formation energy based on density functional theory (DFT). An electronic structure calculation code implemented in the Vienna ab initio simulation package (VASP) with DFT was used, with plane-wave basis sets and Projector-Augmented-Wave (PAW) pseudopotentials under periodic boundary conditions. A Perdew−Burke−Ernzerhof (PBE) functional[62] was used for the exchange correlation with a generalised gradient approximation (GGA). The cut-off energy for wave functions was set at 320 eV, and a $k$-point sampling size of 8 × 8 × 2 was used. The starting structure had four NaKNi$_2$TeO$_6$ formula units (treated as 1 × 1 × 1 supercell).

Full cell relaxation for all the structural configurations (as explicated in Supplementary Figs. 32, 33) were conducted with a tolerance of 10$^{-5}$ eV for total energy of each ion. Spin-polarised GGA+$U$ approach was applied for NaKNi$_2$TeO$_6$ with an antiferromagnetic ordering of Ni wherein the effective Hubbard parameter ($U$) for Ni $d$ orbitals was set at 6 eV.

**X-ray photoelectron spectroscopy (XPS) measurements**. XPS measurements were done on pristine and (dis)charged NaKNi$_2$TeO$_6$ electrodes to ascertain the valency state upon alkali-ion extraction and reinsertion in NaK half-cells. Electrochemical measurements were stopped upon charging the electrodes at 4.35 V and discharging other electrodes at 1.3 V in the first cycle. The cells were dismantled, and the electrodes carefully removed inside an argon-filled glove box (with water and oxygen concentration maintained below 1 ppm). The electrodes were washed for five times using 25 mm of super-dehydrated acetonitrile (water concentration of <10 ppm) and dried inside the argon-filled glove box, prior to undertaking XPS analyses at Te 3$d$ and Ni 2$p$ binding energies. A hermetically sealed vessel was used to transfer the electrode samples into the XPS machine (JEOL(JPS-9030) equipped with a Mg $K\alpha$ source) without exposure to air nor moisture. The electrodes were etched by an Ar-ion beam for 10 s in order to eliminate passivation layer at the surface. The accelerating voltage of Ar-etching was fixed at 600 V. The attained XPS spectra were fitted using Gaussian functions and data processing protocols were performed using COMPRO software.

## Data availability
The data that support the findings of this study are available on request from the corresponding authors.

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

## Acknowledgements

The authors thank Ms. Shinobu Wada and Mr. Hiroshi Kimura for the unrelenting support in undertaking this study. We gratefully acknowledge Ms. Kumi Shiokawa, Mr. Masahiro Hirata and Ms. Machiko Kakiuchi for their advice and technical help as we conducted the syntheses, electrochemical and XRD measurements. This work was supported by the TEPCO Memorial Foundation. In addition, this work was also conducted in part under the auspices of the Japan Society for the Promotion of Science (JSPS KAKENHI Grant Numbers 19K15685 and 21K14730), Sumika Chemical Analyses Services (SCAS) Co. Ltd., National Institute of Advanced Industrial Science and Technology (AIST) and Japan Prize Foundation. We thank Dr. T. Saito for the assistance in the neutron diffraction experiments. The neutron diffraction measurements at J-PARC were performed with the approval of the proposal numbers 2017L1302 and 2020L0802. This study was partly supported by Grants-in-Aid for Scientific Research (KAKENHI, Nos. JP19H00821 and JP20K05086) from the Ministry of Education, Culture, Sports, Science and Technology of Japan. T.Masese. and G.M.K. are grateful for the unwavering support from their family members (T.M.: Ishii Family, Sakaguchi Family and Masese Family; G.M.K.: Ngumbi Family).

## Author contributions

G.M.K. and Z.-D.H. planned the project with T. Masese; T. Masese supervised all aspects of the research with help from G.M.K.; J.R. prepared the honeycomb layered oxide materials with the help from T. Masese; Y.M., M.I. and T.T. and T.S. acquired and analysed TEM data with input from H.S., G.M.K. and T. Masese; J.R., C.–Y.C., K.Y. and T. Masese performed the electrochemical measurements with input from G.M.K., A.A., H.S., J.H., K.M., R.H. and Z.-D.H.; Y.U. and T. Matsunaga acquired the high-resolution X-ray diffraction data. Neutron diffraction data were acquired by M.Y. and K.F. with input from M.S. H.U., C.T. and H.K.; H.U., C.T. and H.K. analysed the neutron diffraction and X-ray diffraction data; T. Matsunaga analysed the X-ray diffraction data using FAULTS program; J.K. performed X-ray photoelectron spectroscopic measurements with input from K.M.; K.K. performed and analysed the thermal stability measurements; T.I. and K.S. performed DFT calculations with input from G.M.K. and T. Masese; The manuscript was written by Y.M., J. R., T.S., C.–Y.C., K.M., R.H., G.M.K., Z.-D.H. and T. Masese, and revised by A.A., H.U., C.T. and H.K. with the help of T. Matsunaga and Y.U. All authors contributed to discussions and made comments pertaining the content in the manuscript and accompanied supplementary material.

## Competing interests

The authors declare no competing interests.
