## [Peer Review File · Nature Communications]

REVIEWER COMMENTS

Reviewer #1 (Remarks to the Author):

In this study, the author demonstrate the feasibility of using a combination of alkali atoms (Na and K) to develop a robust mixed-alkali honeycomb layered oxide $\text{NaKNi}_2\text{TeO}_6$. Through a series of atomic-resolution transmission electron microscopy in multiple zone axes, they reveal for the first time the local atomic structural disorders characterised by aperiodic stackings and incoherency in the alternating arrangement of Na and K atoms. The results not only betoken a new avenue for developing functional materials with fascinating crystal versatility, but also prefigure a new age of 'dendrite-free' energy storage system designs that rely on mixed-cation electrochemistry. Considering the solidness of the manuscript including synthesis, materials characterization, and electrochemical studies, I recommend to publish this work after the following major revisions:

1. In the whole paper, although the structure of the material has been thoroughly studied, the electrochemical properties of the material have been rarely tested. The author should supplement some electrochemical tests to prove your argument.

2. The author should take insights into the electrochemistry behaviors of liquid Na-K alloy batteries, this work is very interesting and meaningful. The liquid anode is dendrite-free and immune to the structural degradations that occur with nanocrystalline electrode particles; therefore, it cycles safely at high rates with excellent capacity retention. But, which element of an alloy anode can reacts first with the electrolyte on discharge and last on charge?

3. Googenought et al (L. Xue, H. Gao, W. Zhou, S. Xin, K. Park, Y. Li, J. B. Goodenough, *Adv Mater* 2016, 28, 9608. L. Xue, H. Gao, Y. Li, J. B. Goodenough, *Journal of the American Chemical Society* 2018, 140, 3292.) have investigated that using of Na/K anode for K ion batteries not only eliminated the metal dendrite growth but also great improved the cycling performance. Xiong's group also reported the insights into the electrochemistry behaviors of solid alkali metal and its liquid alloy in metal batteries (*Small* 2019, 15, 1804916). Those studies have discussed in detail that Na/K metal as the anodes achieved liquid/liquid interphase instead of solid/liquid interphase, which completely solved the dendrite issue and facilitated the ion/electron exchange kinetically.

4. The author said that "When the smaller Na atoms were replaced with larger K atoms (viz., increasing x from 0 to 2), a stepwise shift comprised of several diffraction peaks was observed." So, How do you exactly determine the ratio of Na to K atoms? In other word, how to determine the value of x ?

5. As is see from Figure 4e, f, the voltage-capacity profiles of $\text{NaKNi}_2\text{TeO}_6$ in Na-half cells using 1 M NaFSI in N-methyl-N-propylpyrrolidinium-based (Pyrr13FSI) ionic liquid and (f) K-half cells using 1 M KFSI in Pyrr13FSI ionic liquid. Why use Pyrr13FSI ionic liquid as electrolyte instead of traditional ester or ether electrolyte?

6. Have the authors carried out some post-mortem studies? How about the structural images of the electrode materials after certain number of cycles?

Reviewer #2 (Remarks to the Author):

This is a well executed and interesting study, performed to a high standard. The concept of the mixed alkali, dendrite-free battery is a nice idea, but almost seems like an afterthought to justify the present work. Also no system containing tellurium represents a commercially viable energy storage system.

The electron microscopy is carried out to a high standard, however techniques exist for analysing the XRD data - either by performing a Diffax simulation or more detailed Rietveld analysis using the FAULTS program. One or other of these methods should be employed here.

The electrochemistry measurements are interesting, but why use tungsten foil for the current collector? The density surely reduces the precision of measuring the active mass of material. Also the voltage limits used are particularly wide - the lower cut-off is unnecessarily low. The authors make much of this idea of the mixed alkali dendrite free battery system, but then proceed to bury the data in the supplementary which show that it doesn't actually work as hoped.

Finally the language used in parts of the manuscript is unnecessarily elaborate.

Reviewer #3 (Remarks to the Author):

This paper reports on the mixed-alkali atom-based honeycomb layered oxide (NaK)Ni₂TeO₆ and its characterization using atomic-resolution STEM analysis. The authors reveal interesting local atomic structural disorders of aperiodic stacking and incoherency in the alternating arrangement of Na and K atoms. Specifically, (NaK)Ni₂TeO₆ shows an intriguing alternating arrangement of Na and K atoms sandwiched between Ni_{2/3}Te_{1/3} layers, which was exclusively probed by STEM analysis. This result provides an opportunity to tune the functionality of honeycomb layered oxides by controlling mixed alkali atoms in many applications. However, unlike the STEM analysis, XRD analysis is somehow shallow and needs a more precise and quantitative analysis. Therefore, I can not recommend its publication in the present form. The following issues should be addressed before its publication;

1. While the STEM analysis is excellently executed, the x-ray diffraction analysis is not quantitative and somehow shallow. The authors demonstrated an atomic model structure that accurately describes the observed atomic arrangement by STEM in the atomistic range. However, they failed to describe the long-range crystal structure of (NaK)Ni₂TeO₆ based on the XRD result. As mentioned by the authors (page 6; line 156), newly emerged peaks in diffraction patterns for (NaK)Ni₂TeO₆ are forbidden in the P6₃/mcm symmetry. These new peaks should be emerged by the alternating arrangement of Na and K atoms along the c-direction with an x-ray coherency. Calculation of lattice parameters via profile matching is rather naive. The authors should resolve the detailed crystal structure (long-range) by using their proposed atomistic model, confirmed by the Rietveld refinement. Neutron diffraction analysis is also highly recommended.

2. While only one report on a set of metastable antimonates Li_{3-x}Na_xNi₂SbO₆ has been published (ref 24), the present mixed-alkali atom honeycomb layered oxides (Na_{2-x}K_xNi₂TeO₆) seems stable. What is the origin of this structural stability? Can the authors explain from the thermodynamic point of view? DFT calculation would be very helpful to explain.

3. (page 17; line 442) "This is generally ascribed to the fact that the number of cations participating in extraction and reinsertion would be diminished. As a way to enhance performance, the utilisation of an alloy such as NaK in the case of NaKNi₂TeO₆ would facilitate the participation of both cation species thus yield a realistic theoretical capacity, as demonstrated in Figure S6." => I do not agree with the author's claim that utilizing both cation species (here Na⁺ and K⁺) may yield a realistic theoretical capacity in mixed-alkali battery systems. It is well known that the extraction of whole Na and/or K will collapse the layers, leading to severe capacity fading. The authors need to tone down the statements about the feasibility of mixed-alkali batteries

throughout the manuscript.

4. (Page 9; line 220) "Figure 2a shows a high-angle annular dark-field (HAADF)-STEM image of the parent compound $\text{K}_2\text{Ni}_2\text{TeO}_6$."

=> Figure 2a should be Figure S2a.

Responses to Reviewers' Comments

We greatly appreciate the reviewers for their constructive comments and for the arduous effort and time put into the review of the manuscript. Each comment has been carefully considered, point by point, and responded to. Original reviewers' comments are in boldface, whereas responses are in blue regular typeface. Changes to the text in the manuscript are marked in orange. To facilitate the work of the reviewers, we refer to the revised manuscript indicating, in some instances, the page and the line (page-line). Responses to the reviewers and changes reflected in the revised manuscript are as follows:

Manuscript number: NCOMMS-20-38537-T

Title: Stacking Disorders in Mixed-Alkali Honeycomb Layered Oxide $\text{NaKNi}_2\text{TeO}_6$ and Feasibility for Mixed-Cation Transport

COMMENTS FROM REFEREE #1:

In this study, the authors demonstrate the feasibility of using a combination of alkali atoms (Na and K) to develop a robust mixed-alkali honeycomb layered oxide $\text{NaKNi}_2\text{TeO}_6$. Through a series of atomic-resolution transmission electron microscopy in multiple zone axes, they reveal for the first time the local atomic structural disorders characterised by aperiodic stackings and incoherency in the alternating arrangement of Na and K atoms. The results not only betoken a new avenue for developing functional materials with fascinating crystal versatility, but also prefigure a new age of 'dendrite-free' energy storage system designs that rely on mixed-cation electrochemistry.

Thank you for your thoughtful and thorough review of our manuscript. Benefitting from the revisions suggested, the revised manuscript now has new information and additional interpretations.

Major comments:

Considering the solidness of the manuscript including synthesis, materials characterization, and electrochemical studies, I recommend publishing this work after the following major revisions:

Thank you again for the encouraging comments

1) In the whole paper, although the structure of the material has been thoroughly studied, the electrochemical properties of the material have been rarely tested. The author should supplement some electrochemical tests to prove his argument.

Response: We appreciate your comments greatly. To address the concern raised, we have performed electrochemical measurements to ascertain the mobility of Na⁺ and K⁺ in NaKNi₂TeO₆. Specifically, we have evaluated the electrochemical performance of NaKNi₂TeO₆ in Na half-cells, K half-cells and NaK half-cells, experimental details of which have been appended in the METHODS section. The electrochemical data have also been provided (as Figures 4e, 4f and Figure 6) in the revised manuscript. In addition, we have rewritten the results to elucidate the link between structural aspects of NaKNi₂TeO₆ and the mechanistic aspects demonstrated upon Na / K-ion (de)insertion, as divulged by XRD, XPS and TEM measurements. Please see the XRD, XPS and TEM measurements detailed in comment (6).

2) The author should take insights into the electrochemistry behaviours of liquid Na-K alloy batteries. This work is very interesting and meaningful. The liquid anode is dendrite-free and immune to the structural degradations that occur with nanocrystalline electrode particles; therefore, it cycles safely at high rates with excellent capacity retention. But, which element of an alloy anode reacts first with the electrolyte on discharge and last on charge?

Response: Thank you for the comment. Indeed, battery systems that operate using Na-K liquid alloy are less prone to dendrite formation that often leads to short-circuiting

(‘thermal runaway’) in current rechargeable metal batteries. We have assessed the feasibility of using \$\text{NaKNi}_2\text{TeO}_6\$ mixed-alkali honeycomb layered oxide as a cathode material for NaK battery systems. Alloys encompassing Na and K present rich liquid phase compositions, as shown in **Figure R1-1** below. We utilised the eutectic composition Na_2K as anode (counter electrode) in the assembly of NaK cells. Further details regarding the cell assembly are furnished in the METHODS section.

Figure R1-1. Phase diagram of the Na–K system. A plethora of alloy compositions based on Na and K that are liquid at room temperature can be formed.

Pertaining to the element that takes part in the deposition / dissolution process on the NaK alloy anode side, Na^+ is the first to get out into the electrolyte after which K^+ follow, based on the thermodynamic aspects of Na and K metal deposition and dissolution

(as further shown in **Figure R1-2** below). Note that the metal deposition and dissolution potential of pyrrolidinium-based ionic liquids based on sodium and potassium salts, which are subject of the present study, are shown.

Figure R1-2. Electrochemical stability of TFSI⁻ based ionic liquids containing alkali metal ion at 298 K. Cyclic and linear voltammograms of 0.5 M KTFSI/Py₁₃TFSI and 0.5 M NaTFSI/Py₁₃TFSI are shown, demonstrating the lower potential of metal deposition and dissolution in K relative to Na. For the sake of clarity, the working electrode was a Ni plate. Scan rate was set at 1 mV s⁻¹.

3) Goodenough et al (L. Xue, H. Gao, W. Zhou, S. Xin, K. Park, Y. Li, J. B. Goodenough, *Adv Mater* 2016, 28, 9608. L. Xue, H. Gao, Y. Li, J. B. Goodenough,

Journal of the American Chemical Society 2018, 140, 3292.) have investigated that using of Na/K anode for K ion batteries not only eliminated the metal dendrite growth but also great improved the cycling performance. Xiong's group also reported the insights into the electrochemistry behaviours of solid alkali metal and its liquid alloy in metal batteries (Small 2019, 15, 1804916). Those studies have discussed in detail that Na/K metal as the anodes achieved liquid/liquid interphase instead of solid/liquid interphase, which completely solved the dendrite issue and facilitated the ion/electron exchange kinetically.

Response: Thank you greatly for your suggestion. As recommended by the reviewer, the superiority of NaK metal anode has further been emphasised and we have amply included the recommended bibliographies by the reviewer.

References

1. Goodenough, J. B. *et int.* Liquid K-Na Alloy Anode Enables Dendrite-Free Potassium Batteries. *Adv. Mater.* **28**, 9608 (2016).
2. Goodenough, J. B. *et int.* Cathode Dependence of Liquid-Alloy Na–K Anodes. *J. Am. Chem. Soc.* **140**, 3292 (2018).
3. Xiong, S. *et int.* New Insights into the Electrochemistry Superiority of Liquid Na–K Alloy in Metal Batteries. *Small* **15**, 1804916 (2019).

4) The author said that “When the smaller Na atoms were replaced with larger K atoms (viz., increasing x from 0 to 2), a stepwise shift comprised of several diffraction peaks was observed.” So , How do you exactly determine the ratio of Na to K atoms? In other words, how do you determine the value of x?

Response: Thank you greatly for the comment. SEM-EDX analyses were performed to ascertain the ratio of Na to K in the mixed-alkali honeycomb layered oxide compositions. The SEM-EDX data has been appended in the Supplementary

Information (Supplementary Table S1) for reference. Moreover, the ICP-AES data for the target \$\text{NaKNi}_2\text{TeO}_6\$ have been furnished as Supplementary Table S5. Both the EDX and ICP data univocally indicate the stoichiometry of the as-prepared compositions to be in concordance with the nominal composition x in $\text{Na}_{2-x}\text{K}_x\text{Ni}_2\text{TeO}_6$, as shown in **Figure 1** of the revised manuscript. **Figures R1-3, R1-4, R1-5, R1-6, R1-7, R1-8 and R1-9** show the SEM-EDX results for the as-prepared $\text{Na}_{2-x}\text{K}_x\text{Ni}_2\text{TeO}_6$ ($x = 0, 0.2, 0.6, 1.0, 1.4, 1.8$ and 2.0) compositions, affirming a uniform (homogeneous) distribution of constituent elements. For completeness, the results in Figures R1-3, R1-4, R1-5, R1-6, R1-7, R1-8 and R1-9 below have been appended in the Supplementary Information of our revised manuscript (as Supplementary Figures S2 ~ S8).

Figure R1-3. (a) SEM picture of the $\text{K}_2\text{Ni}_2\text{TeO}_6$ powders. (b) SEM-EDX spectrum of the powders. (c) Elemental mapping of $\text{K}_2\text{Ni}_2\text{TeO}_6$.

Figure R1-4. (a) SEM picture of the $K_{1.8}Na_{0.2}Ni_2TeO_6$ powders. (b) SEM-EDX spectrum of the powders. (c) Elemental mapping of $K_{1.8}Na_{0.2}Ni_2TeO_6$.

Figure R1-5. (a) SEM picture of the $K_{1.4}Na_{0.6}Ni_2TeO_6$ powders. (b) SEM-EDX spectrum of the powders. (c) Elemental mapping of $K_{1.4}Na_{0.6}Ni_2TeO_6$.

Figure R1-6. (a) SEM picture of the $\text{NaKNi}_2\text{TeO}_6$ powders. (b) SEM-EDX spectrum of the powders. (c) Elemental mapping of $\text{NaKNi}_2\text{TeO}_6$.

Figure R1-7. (a) SEM picture of the $\text{Na}_{1.4}\text{K}_{0.6}\text{Ni}_2\text{TeO}_6$ powders. (b) SEM-EDX spectrum of the powders. (c) Elemental mapping of $\text{Na}_{1.4}\text{K}_{0.6}\text{Ni}_2\text{TeO}_6$.

Figure R1-8. (a) SEM picture of the $\text{Na}_{1.8}\text{K}_{0.2}\text{Ni}_2\text{TeO}_6$ powders. (b) SEM-EDX spectrum of the powders. (c) Elemental mapping of $\text{Na}_{1.8}\text{K}_{0.2}\text{Ni}_2\text{TeO}_6$.

Figure R1-9. (a) SEM picture of the $\text{Na}_2\text{Ni}_2\text{TeO}_6$ powders. (b) SEM-EDX spectrum of the powders. (c) Elemental mapping of $\text{Na}_2\text{Ni}_2\text{TeO}_6$.

A quantitative summary of the elemental composition of $\text{Na}_{2-x}\text{K}_x\text{Ni}_2\text{TeO}_6$ ($x = 0, 0.2, 0.6, 1.0, 1.4, 1.8$ and 2.0) mixed-alkali compositions is shown in **Table R1-1** below. The attained compositions are in good agreement with the nominal (expected) compositions. ICP measurements were further conducted for $\text{NaKNi}_2\text{TeO}_6$, as is shown in **Table R1-2**, which is the main focus material composition of this work. We have appended these results in the Supplementary Information of the manuscript.

Table R1-1. Summary of the quantitative elemental composition of $\text{Na}_{2-x}\text{K}_x\text{Ni}_2\text{TeO}_6$ ($x = 0, 0.2, 0.6, 1.0, 1.4, 1.8$ and 2.0) mixed-alkali honeycomb layered oxides.

	atomic weight %				element ratio			
	Na	K	Ni	Te	Na	K	Ni	Te
$\text{Na}_2\text{Ni}_2\text{TeO}_6$	34.72	0.00	44.40	20.88	1.7	0.0	2.1	1.0
$\text{Na}_{1.8}\text{K}_{0.2}\text{Ni}_2\text{TeO}_6$	36.53	4.45	37.76	21.27	1.7	0.2	1.8	1.0
$\text{Na}_{1.4}\text{K}_{0.6}\text{Ni}_2\text{TeO}_6$	27.58	10.15	41.94	20.34	1.4	0.5	2.1	1.0
$\text{NaKNi}_2\text{TeO}_6$	18.98	18.00	42.41	20.60	0.9	0.9	2.1	1.0
$\text{Na}_{0.6}\text{K}_{1.4}\text{Ni}_2\text{TeO}_6$	11.55	26.73	40.81	20.91	0.6	1.3	2.0	1.0
$\text{Na}_{0.2}\text{K}_{1.8}\text{Ni}_2\text{TeO}_6$	4.03	33.89	41.73	20.35	0.2	1.7	2.1	1.0
$\text{K}_2\text{Ni}_2\text{TeO}_6$	0.48	38.12	41.37	20.03	0.0	1.9	2.1	1.0

We noted a propensity to prepare highly stoichiometric compositions using slightly an excess amount of alkali carbonates.

Table R1-2. Composition stoichiometry of constituent Na, K, Ni and Te in pristine $\text{NaKNi}_2\text{TeO}_6$. Table showing the experimental stoichiometry values obtained from inductively coupled plasma atomic emission spectroscopy (ICP-AES) measurement of pristine $\text{NaKNi}_2\text{TeO}_6$ synthesised at 800°C in this study.

compound	Na (mole ratio)	K (mole ratio)	Ni (mole ratio)	Te (mole ratio)
$\text{NaKNi}_2\text{TeO}_6$	0.999(± 0.001)	0.995(± 0.005)	2.01(± 0.01)	1

These results have been added as Supplementary Information of our revised manuscript (Supplementary Tables S1 and S5).

5) As is seen from Figure 4e and 4f, the authors display the voltage-capacity profiles of NaK₂Ni₂TeO₆ in (e) Na-half cells using 1 M NaFSI in N-methyl-N-propylpyrrolidinium-based (Pyrr₁₃FSI) ionic liquid and (f) K-half cells using 1 M KFSI in Pyrr₁₃FSI ionic liquid. Why use Pyrr₁₃FSI ionic liquid as electrolyte instead of traditional ester or ether electrolyte?

Response: Thank you greatly for giving us a chance to clarify this point. It is well-documented that polarisation (overpotential) issues of Na and K metal deposition–dissolution in conventional organic electrolytes (traditional ester- or ether-based electrolytes) can profoundly influence the electrochemical performance obtained in half-cell configurations.^{1,2} Bearing this in mind, ionic liquid electrolytes were adopted because Na and K metal deposition–dissolution behaviour has been extensively studied and found to proceed with acceptably low overpotential.³⁻⁵

In addition, ether- and ester-based solvents are known for their limited oxidative stability (< 4.0 V and < 4.5 V versus Li⁺/Li, respectively).⁶⁻⁸ Electrolytes consisting of ionic liquids, *vide infra*, exhibit higher oxidative stability (> 5.0 V versus Na⁺/Na or K⁺/K)⁵; thus aid to more accurately evaluate the electrochemical performance of high-voltage cathode materials such as the mixed-alkali honeycomb layered oxide NaK₂Ni₂TeO₆ at high-voltage regimes. The nonflammability of ionic liquids further makes the battery intrinsically safe, particularly when using alkali metal as anodes (counter electrodes).³

Therefore, we utilised pyrrolidinium-based ionic liquids based on NaFSI and KFSI in performing electrochemical tests in Na half-cells and K half-cells, respectively. The physicochemical properties of these ionic liquids and their compatibility with high-voltage honeycomb layered oxide cathode materials have been reported in our previous works.⁹⁻¹¹ The physicochemical properties of a new mixed ionic liquid (0.5 M NaFSI + 0.5 M

KFSI in Pyr₁₃FSI) used, in the present study, as an electrolyte for the NaK half-cell are shown in **Table R1-3** below. Although the viscosity is relatively high, considerably good ionic conductivity is attainable at room temperature warranting its utilisation as a suitable electrolyte for electrochemical evaluation of electrodes in NaK cells.

Table R1-3. Physicochemical properties of a mixed ionic liquid comprising a 0.5 M NaFSI + 0.5 M KFSI in Pyr₁₃FSI. Data for 1 M KFSI in Pyr₁₃FSI and 1 M NaFSI in Pyr₁₃FSI ionic liquid electrolytes have been provided for a comparison. Measurements were performed at room temperature (25°C).

Ionic liquids	Density / g cm ⁻³	Viscosity / mPa s	Ionic conductivity / mS cm ⁻¹
1.0 M KFSI/Pyr ₁₃ FSI	1.4226	83.9	4.4
1.0 M NaFSI/Pyr ₁₃ FSI	1.4119	94.9	4.1
0.5 M NaFSI + 0.5 M KFSI/ Pyr ₁₃ FSI	1.4232	89.2	4.3

For clarity to the readers, we have explicitly stated why we utilised electrolytes based on ionic liquids in the METHODS section. This table has been further appended in the Supplementary Information section (Supplementary Table S6).

Reference

1. Conder, J. & Villevieille, C. How reliable is the Na metal as a counter electrode in Na-ion half cells. *Chem. Commun.* **55**, 1275–1278 (2019).
2. Komaba, S. *et al.* Potassium metal as reliable reference electrodes of nonaqueous potassium cells. *J. Phys. Chem. Lett.* **10**, 3296–3300 (2019).
3. Matsumoto, K. *et al.* Advances in sodium secondary batteries utilizing ionic liquid electrolytes. *Energy Environ. Sci.* **12**, 3247–3287 (2019).
4. Hwang, J., Matsumoto, K. & Hagiwara, R. Symmetric cell electrochemical impedance spectroscopy of Na₂FeP₂O₇ positive electrode material in ionic liquid electrolytes. *J. Phys. Chem. C* **122**, 26857–26864 (2018).
5. Yamamoto, T., Matsumoto, K., Hagiwara, R. & Nohira, T. Physicochemical and electrochemical properties of K[N(SO₂F)₂]-[N-methyl-N-propylpyrrolidinium][N(SO₂F)₂] ionic liquid for potassium-ion batteries. *J. Phys. Chem. C* **121**, 18450–18458 (2017).

6. Jiao, S. H. *et al.* Stable cycling of high-voltage lithium metal batteries in ether electrolytes. *Nat. Energy* **3**, 739–746 (2018).
7. Jie, Y. L. *et al.* Enabling high-voltage lithium metal batteries by manipulating solvation structure in ester electrolyte. *Angew. Chem. Int. Ed.* **59**, 3505–3510 (2020).
8. Xu, K. Electrolytes and interphases in Li-ion batteries and beyond. *Chem. Rev.* **114**, 11503–11618 (2014).
9. Yoshii, K. *et al.* Sulfonamide - Based Ionic Liquids for High - Voltage Potassium - Ion Batteries with Honeycomb Layered Cathode Oxides. *ChemElectroChem.* **6**, 3901–3910 (2019).
10. Chen, C. -Y. *et al.* High-voltage honeycomb layered oxide positive electrodes for rechargeable sodium batteries. *Chem. Commun.* **56**, 9272–9275 (2020).
11. Kanyolo, G. M. *et al.* Honeycomb layered oxides: structure, energy storage, transport, topology and relevant insights. *Chem. Soc. Rev.* in press (2021). <https://doi.org/10.1039/D0CS00320D>

6) Have the authors carried out some post-mortem studies? How about the structural images of the electrode materials after certain number of cycles?

Response: The reviewer has raised an important aspect that is worth investigating. We have now performed TEM, XRD and XPS measurements on discharged NaKNi₂TeO₆ electrodes upon subsequent cycling in Na, K and NaK half-cells. The measurement protocols and results have been added to RESULTS and METHODS section of our revised manuscript (page 22, line 11-13 and Supplementary Figure S22). For lucidity, we explain the various approaches made to discriminate the structural changes in the following subsections **(a)**, **(b)**, **(c)**, **(d)** and **(e)**.

(a) Cycling in Na half-cells (high resolution TEM measurements)

Figure R1-10 shows high-resolution STEM micrographs taken along the [100] axis of NaKNi₂TeO₆ upon subsequent cycling in Na half-cells. Details pertaining to the

measurement are furnished in the **METHODS** section of the revised manuscript. $\text{NaKNi}_2\text{TeO}_6$ transforms to $\text{Na}_2\text{Ni}_2\text{TeO}_6$ upon successive cycling (in this case, after three cycles) in Na half-cells. The voltage-capacity plots of $\text{NaKNi}_2\text{TeO}_6$ in Na half-cells (shown in **Figure 4c** in the revised manuscript) reveal a reversible capacity of ca. 80 mAh g^{-1} attainable upon subsequent cycling. The theoretical capacity for a full Na^+ extraction from $\text{NaKNi}_2\text{TeO}_6$ is approximately 67 mAh g^{-1} , suggesting the extraction of K^+ which are successively replaced by Na^+ during the reinsertion process to yield $\text{Na}_2\text{Ni}_2\text{TeO}_6$ as affirmed by the high-resolution TEM measurements shown in **Figure R1-10**.

Figure R1-10. Structural changes during alkali-ion (de)insertion of $\text{NaKNi}_2\text{TeO}_6$ upon subsequent cycling in Na half-cells. (a) HAADF-STEM images of $\text{NaKNi}_2\text{TeO}_6$ taken along the [100] axis upon cycling in a Na half-cell. (b) Corresponding ABF-STEM images and (c) SAED patterns showing streaks (shown in green arrows)

indicative of the modulation in the Na arrangement along the *ab* plane as is exemplified in Na₂Ni₂TeO₆.¹ (d) Intensity line profiles as highlighted in (a) showing equidistant interlayer spacings (0.55 nm) of Na atoms along the [001] axis as quantitatively illustrated in (e). For clarity, the horizontal axis in (e) shows the layer numbers.

These results have been added to RESULTS section of our revised manuscript (page 21, revised Figures 5a, 5b, 5c and 5d).

Reference

1. Masese, T. *et al.*, Unveiling structural disorders in honeycomb layered oxide: Na₂Ni₂TeO₆. *Materialia*. **15**, 101003 (2021).

(b) Cycling in K half-cells (high resolution TEM measurements)

Voltage-capacity plots of NaKNi₂TeO₆ upon subsequent cycling in K half-cells reveal a reversible initial capacity of approximately 50 mAh g⁻¹ (shown in **Figure 4d** in the revised manuscript). The theoretical capacity for a full Na⁺ extraction from NaKNi₂TeO₆ is approximately 67 mAh g⁻¹, which indicates that the capacity arises predominantly from K⁺ extraction. Given that K metal was used as anode (counter electrode), reversible extraction and reinsertion of K can be envisaged. To test this hypothesis, STEM measurements were performed (as shown in **Figure R1-11** below) revealing that the mixed alkali layered framework is retained owing to reversible extraction and insertion of K alone.

Figure R1-11. Structural changes during alkali-ion (de)insertion of $\text{NaKNi}_2\text{TeO}_6$ upon subsequent cycling in K half-cells. (a) HAADF-STEM images of $\text{NaKNi}_2\text{TeO}_6$ taken along the [100] axis upon cycling in a K half-cell. (b) Corresponding ABF-STEM images and (c) SAED patterns showing streaks indicative of the aperiodic stacking nature along the c -axis. (d) Intensity line profiles as highlighted in (a) showing alternating interlayer spacings (0.55 nm and 0.62 nm) of Na and K atoms, respectively, along the [001] axis. This is further quantitatively illustrated in (e). For clarity, the horizontal axis in (e) shows the layer numbers.

These results have been added to RESULTS section of our revised manuscript (page 21, revised Figures 5e, 5f, 5g and 5h).

(c) Cycling in NaK half-cells (high resolution TEM measurements)

Voltage-capacity plots of $\text{NaKNi}_2\text{TeO}_6$ when cycled in a NaK half-cell, using a mixed electrolyte based on a 0.5 M NaFSI + 0.5 M KFSI in $\text{Pyr}_{13}\text{FSI}$ ionic liquid, displays a reversible capacity of about 80 mAh g^{-1} . Since both Na and K reinsertion can be envisaged, further structural insights were attained from high-resolution STEM measurements (as is furnished in **Figure R1-12**). Domains with Na-rich phases ($\text{Na}_2\text{Ni}_2\text{TeO}_6$) were found in the crystallites analysed. This indicates that Na^+ are predominantly inserted into the layered framework upon successive cycling. We can rationalise these findings from the fact that Na^+ deposition and dissolution (stripping) occur first at the anode, owing to the thermodynamic potential of Na being higher than for K. Moreover, recent reports have revealed that the alkali-metal stripped from the NaK anode side is dependent on the preference of the cathode.¹ The fact that the cathode prefers Na^+ to K^+ (**Figure R1-12**), hints that Na^+ are stripped and deposited first at the anode as should be expected.

Reference

1. Goodenough, J. B. *et al.* Cathode Dependence of Liquid-Alloy Na–K Anodes. *J. Am. Chem. Soc.* **140**, 3292 (2018).

Figure R1-12. Structural changes during alkali-ion (de)insertion of NaKNi₂TeO₆ upon subsequent cycling in NaK half-cells. HAADF-STEM images of NaKNi₂TeO₆ upon cycling in NaK battery cell taken along (a) [100] axis and (b) [1 $\bar{1}$ 0] axis. Corresponding ABF-STEM images are furnished in the Supplementary Information section. (c) Intensity line profiles as highlighted in (a) showing equidistant interlayer spacings (0.55 nm) along the [001] axis revealing NaKNi₂TeO₆ to

preferentially insert Na atoms into the lattice. This is further quantitatively illustrated in (d). For clarity, the horizontal axis in (d) shows the layer numbers.

These results have been added to RESULTS section of our revised manuscript (page 25, revised Figures 7a, 7b and 7c).

Figure R1-13 below further shows the SAED patterns of $\text{NaKNi}_2\text{TeO}_6$ upon subsequent cycling in NaK half-cells. Streaks are apparent in both the $[100]$ and $[1\bar{1}0]$ zone axes, indicative of the highly ordered double periodicity used to describe the arrangement of Na atoms in $\text{Na}_2\text{Ni}_2\text{TeO}_6$ along the ab plane. More details relating to the $2a \times 2a$ superstructure of Na site occupancy in the ab plane of $\text{Na}_2\text{Ni}_2\text{TeO}_6$ are provided in Figure R1-14, for clarity.

Figure R1-13. Structural changes during alkali-ion (de)insertion of $\text{NaKNi}_2\text{TeO}_6$ upon subsequent cycling in NaK half-cells. SAED patterns taken along (a) $[100]$ axis and (b) $[1\bar{1}0]$ axis showing streaks (underpinned in green arrows) indicative of the modulation in the Na arrangement as is exemplified in $\text{Na}_2\text{Ni}_2\text{TeO}_6$.

These results have been added to RESULTS section of our revised manuscript (page 25, revised Figures 7d and 7e).

Figure R1-14.¹ Comparison of the electron diffractograms of $\text{Na}_2\text{Ni}_2\text{TeO}_6$ based on the original cell and the double-periodicity supercell. (a) Selected area electron diffraction (SAED) patterns of $\text{Na}_2\text{Ni}_2\text{TeO}_6$ taken along the $[100]$ zone axis highlighting streaks (green arrows) in the diffractograms hallmarking the presence of a supercell. (b) Simulated diffractograms along the same $[100]$ axis using the original cell and (c) Supercell with a manifold dimensions of the unit cell along the a -axis and b -axis, which reproduces the experimentally obtained electron diffraction patterns of $\text{Na}_2\text{Ni}_2\text{TeO}_6$. (d) SAED patterns of $\text{Na}_2\text{Ni}_2\text{TeO}_6$ taken along the

$[1 \bar{1} 0]$ axis also underpinning streaks in the diffractograms. (e) Simulated diffractograms along the same $[100]$ axis using the original cell and (f) Supercell with a manifold dimensions of the unit cell along the a -axis and b -axis, which reproduces the experimentally obtained electron diffraction patterns. (g) Arrangement of the Na site occupancy within the ab plane in $\text{Na}_2\text{Ni}_2\text{TeO}_6$ based on a $a \times a$ unit cell and a double periodicity $2a \times 2a$ supercell.

Reference

1. Masese, T. *et al.*, Unveiling structural disorders in honeycomb layered oxide: $\text{Na}_2\text{Ni}_2\text{TeO}_6$. *Materialia*. **15**, 101003 (2021).

(d) Cycling in Na, K and NaK half-cells (*ex situ* XRD measurements)

XRD measurements were also performed on electrodes upon subsequent cycling in Na, K and NaK half-cells, in order to affirm the conclusions drawn from TEM measurements. **Figure R1-15** below shows the *ex situ* XRD measurements for discharged composite electrodes. Details regarding the measurement protocols are furnished in the METHODS section of the revised manuscript.

Successive cycling of $\text{NaKNi}_2\text{TeO}_6$ in Na half-cells yields the Na-rich phase $\text{Na}_2\text{Ni}_2\text{TeO}_6$, whilst mixed alkali phase ($\text{NaKNi}_2\text{TeO}_6$) is principally maintained in a K half-cell. This is in accord with what was observed in TEM measurements.

Subsequent cycling of $\text{NaKNi}_2\text{TeO}_6$ in a NaK cell yields the Na-rich phase ($\text{Na}_2\text{Ni}_2\text{TeO}_6$) as the predominant phase, albeit there is a remanent mixed alkali $\text{NaKNi}_2\text{TeO}_6$ and K-rich phase ($\text{K}_2\text{Ni}_2\text{TeO}_6$). The crystallites tested using TEM reveal principally the Na-rich phases. Although the other minor phases could not be discerned at the local scale in the crystallites tested, the XRD results further suggest that there are still some remaining Na-lean phases that presumably transform to the Na-rich phase upon prolonged cycling.

Figure R1-15. Crystal structural changes during alkali-ion (de)insertion of $\text{NaKNi}_2\text{TeO}_6$ upon subsequent cycling in Na, K and NaK half-cells based on XRD measurements.

These results have been added to RESULTS section of our revised manuscript (page 24, line 26-30, Supplementary Figure S24).

Further nanostructural details relating to the Na-rich phases reveal stacking disorders along the *c*-axis, as is schematically shown in **Figure R1-16**, entailing shifts/shear of the transition metal slabs along the *ab* plane by a Burgers vector corresponding to $[\pm 2/3 \pm 1/3 0]$.

Figure R1-16. Stacking arrangement of the Na-rich phase $\text{Na}_2\text{Ni}_2\text{TeO}_6$ attained upon subsequent cycling of $\text{NaKNi}_2\text{TeO}_6$ in NaK half-cells.

These results have been added as Supplementary Information of our revised manuscript (Supplementary Note 1).

The slabs in the Na-rich phase ($\text{Na}_2\text{Ni}_2\text{TeO}_6$) are observed to deviate from the vertical arrays in certain domains (as highlighted by the green lines in **Figure R1-16**). For the sake of rigour, a shear transformation is defined by matrices of the type,

$$S_a = \begin{pmatrix} 1 & 0 & \lambda_a \\ 0 & 1 & 0 \\ 0 & 0 & 1 \end{pmatrix}, S_b = \begin{pmatrix} 1 & 0 & 0 \\ 0 & 1 & \lambda_b \\ 0 & 0 & 1 \end{pmatrix},$$

where S_a and S_b are the shear matrices in the a and b directions respectively. Due to the differing crystalline positions of Ni atoms relative to Te atoms in the unit cell at the stacking faults, the Burgers vector corresponds to $\lambda_a = \pm 2/3$ and $\lambda_b = \pm 1/3$ anisotropic shear transformations along the ab plane. Equivalently, the shear transformations lead to a Ni atom exchanging relative positions with a Te atom within the unit cell. This requires that the unit basis vector perpendicular to the ab plane (pointing in the c axis) given by the transpose of the basis vector $[0, 0, 1]$ be multiplied by the combined transformation matrix $S = S_a S_b = S_b S_a$ given by,

$$S = \begin{pmatrix} 1 & 0 & \lambda_a \\ 0 & 1 & \lambda_b \\ 0 & 0 & 1 \end{pmatrix},$$

to yield, $[\pm 2/3, \pm 1/3, 1]$ but leaves unchanged the transpose basis vectors in the a - and b -axes *i.e.* $[1, 0, 0]$ and $[0, 1, 0]$ vectors respectively. Correspondingly, the Burgers vector $[\pm 2/3, \pm 1/3, 0]$ is the difference between the transformed basis vector $[\pm 2/3, \pm 1/3, 1]$ and the original unit basis vector $[0, 0, 1]$ as shown in the **Rendition R1** below:

Rendition R1: The basis vectors defined on the unit cell relative to the shear transformation. (a) The c -zone axis unit basis vector $[0, 0, 1]$, its transformed vector $[\pm 2/3, \pm 1/3, 1]$ under shear transformation, S and the corresponding Burgers vector $[\pm 2/3, \pm 1/3, 0]$ depicting the Te ion occupying the relative position of Ni. (b) The shear transformation in (a) as seen from the $[001]$ axis (ab plane) showing the unit basis vectors $[0, 0, 1]$, $[0, 1, 0]$ and $[1, 0, 0]$, and the Burgers vector, $[\pm 2/3, \pm 1/3, 0]$. (Provided as Supplementary Note 1).

(e) Cycling in NaK half-cells (*ex situ* XPS measurements)

Having garnered crystal-structural insights based on high resolution TEM and XRD measurements, we were compelled to further investigate the electronic structural changes in order to attain a holistic picture of the structural changes occurring in NaKNi₂TeO₆. To understand the charge compensation process occurring in NaKNi₂TeO₆ during alkali-ion extraction and reinsertion in a NaK cell, the evolution of the chemical composition and the transition metal oxidation state of the pristine, charged, and discharged electrodes in NaK cells were examined by *ex situ* XPS measurements (see **Figure R1-17**).

Looking at the Te 3*d* core spectra (**Figure R1-17a**), the Te 3*d*_{5/2} peak centered at around 576.0 eV (which is characteristic of Te⁶⁺) remains invariant for the entire charge–discharge process. This indicates that Te, as a spectator ion, does not participate in the redox process. This has also been observed upon alkali-ion extraction and reinsertion in related honeycomb layered oxides such as K₂Ni₂TeO₆ upon charging and discharging.¹ Turning to the Ni 2*p* core spectra (**Figure R1-17b**), *vide infra*, changes are discernible indicative of the participation of the Ni 2*p* orbitals in the charge compensation process. Although the Ni³⁺ at the electrode surface obfuscates the complete capture of the Ni²⁺ peak signal, a reversible oxidation of Ni²⁺ during charging and a reduction of Ni³⁺ during discharging is evident. Ni²⁺ is evinced by the main peak at 854.0 eV whereas the peak at 861.6 eV emanates from the auger spectra of Ni 2*p*_{3/2} (denoted as satellite peaks). The binding energy of the Ni 2*p*_{3/2} peak shifts to a higher value of 856.4 eV, which corresponds to Ni³⁺ after charging. Upon discharging, the Ni 2*p*_{3/2} peak signal can be gleaned, confirming that Ni 2*p* takes part in the charge compensation process. Although the bulk-sensitive XAS measurements are beyond the scope of the current work, the charge compensation process in NaKNi₂TeO₆ can be envisioned to be similar to that reported in related nickel tellurates such as K₂Ni₂TeO₆.¹

Figure R1-17. Electronic structural changes during alkali-ion (de)insertion of NaKNi₂TeO₆ upon subsequent cycling in NaK half-cells. (a) Te 3d and (b) Ni 2p XPS spectra taken for the pristine, charged and discharged electrodes. The satellite peaks are marked as ‘Ni sat.’.

These results have been added to RESULTS section of our revised manuscript (page 22, line 31-35, Supplementary Figure S22).

Reference

1. Masese, T. *et al.*, Rechargeable potassium-ion batteries with honeycomb-layered tellurates as high voltage cathodes and fast potassium-ion conductors. *Nat. Commun.* **9**, 1-12 (2018).

COMMENTS FROM REFEREE #2:

This is a well-executed and interesting study, performed to a high standard. The concept of the mixed alkali, dendrite-free battery is a nice idea, but almost seems like an afterthought to justify the present work. Also, no system containing tellurium represents a commercially viable energy storage system.

The electron microscopy is carried out to a high standard; however, techniques exist for analysing the XRD data - either by performing a Diffax simulation or more detailed Rietveld analysis using the FAULTS program. One or other of these methods should be employed here.

The electrochemistry measurements are interesting, but why use tungsten foil for the current collector? The density surely reduces the precision of measuring the active mass of material. Also, the voltage limits used are particularly wide - the lower cut-off is unnecessarily low. The authors make much of this idea of the mixed alkali dendrite free battery system, but then proceed to bury the data in the supplementary which show that it does not actually work as hoped.

Finally, the language used in parts of the manuscript is unnecessarily elaborate.

We are greatly encouraged by the reviewer's overall assessment of our manuscript and for the specific comments to improve our paper. We have modified the discussion in the manuscript to clarify the points based on the reviewer's suggestions. The detailed response to each point raised by the reviewer and the related modification in the revised manuscript are as follows.

1) The concept of the mixed alkali, dendrite-free battery is a nice idea, but almost seems like an afterthought to justify the present work.

Response: We thank the reviewer for giving us a chance to clarify this point. The concept of using $\text{NaKNi}_2\text{TeO}_6$ as a potential cathode material for NaK batteries was not fortuitous. We succinctly explain our rationale hereafter.

Batteries based on NaK alloy anodes indeed present an energy storage system that does not face the risk of alkali metal dendrite formation.¹⁻⁴ Currently, a few cathode candidates have been reported all of which contain only Na or K in their framework (e.g. $\text{Na}_2\text{MnFe}(\text{CN})_6$, $\text{K}_{0.22}\text{Fe}[\text{Fe}(\text{CN})_6]$).⁵⁻¹⁰ However, to date, no cathode material has been reported that incorporates both Na and K initially in its framework. $\text{NaKNi}_2\text{TeO}_6$ is the first of its kind to contain both Na and K in its initial framework.

High-resolution transmission electron microscopy (as shown in **Figure R2-1** below) reveals Na and K atoms to be capriciously aligned in an alternating manner in the as-prepared $\text{NaKNi}_2\text{TeO}_6$.

Figure R2-1. Arrangement of atoms in $\text{NaKNi}_2\text{TeO}_6$, viewed along the $[001]$ and $[1\bar{1}0]$ zone axes. STEM images of $\text{NaKNi}_2\text{TeO}_6$ taken along the $[100]$ and $[1\bar{1}0]$ zone axes.

(a) High-angle annular dark-field (HAADF) STEM (HAADF-STEM) images taken along the [100] illustrating the alternating arrangement stacking sequence of Na and K atoms in $\text{NaKNi}_2\text{TeO}_6$, based on the analyses of the corresponding annular bright-field (ABF)-STEM images shown in (b). (c) HAADF-STEM images showing the alternating arrangement of K and Na layers. The bright spots in the HAADF-STEM image correspond to Te and Ni atoms residing in the honeycomb slabs. (d) Corresponding ABF-STEM images where the Te and Ni atoms are indicated by the dark spots. The Na, and K atoms can also be observed as light grey sections located between Te/Ni slabs.

These results have been provided in Figure 2g of the revised manuscript and in the Supplementary Information section (Supplementary Figure S13).

Moreover, molecular dynamics (MD) simulations indicate that the Na and K are mobile (as furnished in the **Supplementary Video**). Therefore, this served as an impetus for us to assess the feasibility of utilising $\text{NaKNi}_2\text{TeO}_6$ as a potential cathode material for NaK batteries.

To address the concern raised, we have performed electrochemical measurements to justify $\text{NaKNi}_2\text{TeO}_6$ to be electrochemically active not only in Na and K half-cells, but also in NaK half-cells. The electrochemical data has been provided in **Figures 4** and **6** of the revised manuscript. We have also performed TEM and XRD measurements on discharged $\text{NaKNi}_2\text{TeO}_6$ electrodes to ascertain the structural changes during successive cycling (shown in **Figures 5** and **7** of the revised manuscript). Details regarding experimental setup have been furnished in the **METHODS** section. Further, a discussion on the electrochemical aspects of $\text{NaKNi}_2\text{TeO}_6$ has been made in the **DISCUSSION** section of the revised manuscript.

Reference

1. Goodenough, J. B. *et al.* Liquid K-Na Alloy Anode Enables Dendrite-Free Potassium Batteries. *Adv. Mater.* **28**, 9608 (2016).

2. Goodenough, J. B. *et al.* Cathode Dependence of Liquid-Alloy Na–K Anodes. *J. Am. Chem. Soc.* **140**, 3292 (2018).
3. Xiong, S. *et al.* New Insights into the Electrochemistry Superiority of Liquid Na–K Alloy in Metal Batteries. *Small* **15**, 1804916 (2019).
4. Chen, C. -Y. *et al.* An Energy - Dense Solvent - Free Dual - Ion Battery. *Adv. Funct. Mater.* **30**, 2003557 (2020).
5. Xue, L. G., Gao, H. C., Li, Y. T. & Goodenough, J. B. Cathode dependence of liquid-alloy Na-K anodes. *J. Am. Chem. Soc.* **140**, 3292–3298 (2018).
6. Zhang, L. Y., Xia, X. H., Zhong, Y., Xie, D., Liu, S. F., Wang, X. L. & Tu, J. P. Exploring self-healing liquid Na-K alloy for dendrite-free electrochemical energy storage. *Adv. Mater.* **30**, 1804011 (2018).
7. Ding, Y., Guo, X. L. Qian, Y. M. Zhang, L. Y. Xue, L. G., Goodenough, J. B. & Yu, G. H. A liquid-metal-enabled versatile organic alkali-ion battery. *Adv. Mater.* **31**, 1806956 (2019).
8. Kapaev, R. R., Obrezkov, F. A., Stevenson, K. J. & Troshin, P. A. Metal-ion batteries meet supercapacitors: high capacity and high rate capability rechargeable batteries with organic cathode and a Na/K alloy anode. *Chem. Commun.* **55**, 11758–11761 (2019).
9. Huang, M., Xi, B. J., Feng, Z. Y., Wu, F. F., Wei, D. H., Liu, J. Feng, J. K., Qian, Y. T. & Xiong, S. L. New insights into the electrochemistry superiority of liquid Na-K alloy in metal batteries. *Small* **15**, 1804916 (2019).
10. Ding, Y., Guo, X. L., Qian, Y. M., Xue, L. G., Dolocan, A. & Yu, G. H. Room-temperature all-liquid-metal batteries based on fusible alloys with regulated interfacial chemistry and wetting. *Adv. Mater.* **32**, 2002577 (2020).

2) No system containing tellurium represents a commercially viable energy storage system.

Response: We thank the reviewer for giving us a chance to clarify this point. We indeed recognise, *sensu stricto*, that tellurium is not a constituent element of choice for energy

storage systems entailing Earth-abundant elements, such as the nascent NaK batteries, mainly due to the high price of tellurium in the present market. However, we do not think this is a good enough reason not to study the rudimentary properties and feasibility of these materials since the insights herein are not exclusive to tellurates. Moreover, one cannot rule out the use of tellurates in niche applications where functionality may be prioritised over cost. Nonetheless, we present the present work as a **fundamental study** of the NaK tellurates. This point has now been underscored in the **DISCUSSION** section (page 19, line 18-line 20).

Currently, there is an **uptick in demand for tellurium** driven by new applications that use high-purity Te. We note that tellurium (a chalcogen) has several important commercial uses. It is primarily used for manufacturing films essential to photovoltaic solar cells as well as thermoelectric applications. For instance, when alloyed with cadmium – the leading end-use amongst these applications cadmium telluride-based solar cells can be produced.¹ Tellurium is further used in copying machines and as a colouring agent in ceramics and glass, and as a vulcanising agent in the chemical industry to make durable products, including an additive that improves rubber's heat resistance.² Moreover, medical instrumentation, integrated circuits, and laser diodes, all of which have experienced robust manufacturing growth in recent years, contain tellurium.^{3,4} In addition, lead and germanium telluride-based materials, which display intriguing functionalities, have also been intensively studied from both fundamental and technological perspectives.^{5,6} To reiterate, although **tellurium may not be a terrigenous element**, it has intensively been studied from both **fundamental and technological perspectives**, owing to the intriguing functionalities endowed in tellurium-based compounds

On a **materials cost perspective**, the content of tellurium used in the tellurate-based mixed-alkali $\text{NaKNi}_2\text{TeO}_6$ material is relatively low to significantly influence the total material cost of preparation, if it were to be utilised as a functional material (for instance, battery material). Preliminary material cost calculations for honeycomb layered tellurates such as $\text{NaKNi}_2\text{TeO}_6$ reveal 1 g to cost around ¥54 (translating to \$0.51 or thereabout)

which is close to 50% reduction in the material costs, in comparison to lithium battery cathode materials (for example) such as the stellar LiCoO_2 or $\text{LiNi}_{0.8}\text{Co}_{0.1}\text{Co}_{0.1}\text{O}_2$ ($\sim\text{¥}110$ ($\sim\text{\$}1$)/g). Thus, the presence of tellurium is not a deterrent to the total material cost. With bespoke manufacturing (synthesis) protocols and greater economies of scale (for instance, as governed by well-established market cost parameters such as the Herfindahl-Hirschman index), honeycomb layered tellurates can be envisaged to be prepared on pilot scale at affordable costs for various applications. Furthermore, the packing casings of batteries are the main components that significantly influence the cost/price of a battery pack.⁷

This study underpins $\text{NaKNi}_2\text{TeO}_6$ as a high-voltage cathode contender for the state-of-the-art NaK batteries (concept of which is shown in **Figure R2-2a**), comprising Na and K capriciously aligned in alternating configuration within its layered crystal framework. We have shown that $\text{NaKNi}_2\text{TeO}_6$ is amenable to reversible (de)insertion of both Na^+ and K^+ (de)insertion at high-voltages and with decent capacities, which is a crucial leap in NaK battery systems where there is still a dearth of high-energy-density cathode materials. This is apparent in **Figure R2-2b** below which univocally shows the honeycomb layered mixed Na/K tellurates reigning supreme as high voltage cathode material contenders. A compendium of the cathode materials reported so far for rechargeable NaK batteries is also provided in **Table R2-1**. We reiterate that, from a fundamental standpoint, this study provides valuable information in the hunt for even better cathode materials for rechargeable batteries operating on NaK liquid metal alloy as anode.

Figure R2-2. (a) A schematic illustration of a dual (Na- and K-ion) battery using $\text{NaKNi}_2\text{TeO}_6$ as cathode and dendrite-free NaK liquid alloy as anode. (b) Comparison of the average voltage, specific capacity, and energy density of

cathode materials reported for batteries relying on NaK liquid metal alloy as anode.

¹⁻⁶ The filled and hollow circles indicate the practically achieved and theoretical capacities, respectively. The energy density is calculated based on the weight of cathode material (Please see Table R2-1 below for details).

These results have been provided in the revised manuscript (Figures 6a and 6e).

Table R2-1. Electrochemical properties of positive electrode (cathode) materials reported for NaK anodes.

This table has been added to as Supplementary Information (Supplementary Table S4) of our revised manuscript.

Positive electrode material	Negative electrode material	Electrolyte	Average voltage / V	Theoretical capacity ^{a)} / mAh g ⁻¹	Achieved capacity / mAh g ⁻¹	Remarks	Ref.
NaKNi ₂ TeO ₆	NaK	0.5 M NaFSI + 0.5 M KFSI in Pyr ₁₃ FSI	3.8	133	82	Layered structured (P6 ₃ /mcm)	This work
	Na	1.0 M NaFSI in Pyr ₁₃ FSI	3.8		86	First inorganic stable host containing both Na and K	
	K	1.0 M KFSI in Pyr ₁₃ FSI	3.9		52	Highest average voltage	
Na ₂ MnFe(CN) ₆	NaK	1.0 M NaClO ₄ in PC	3.4	171	~120	Prussian blue analogue with preference to K ions (transformed to K ₂ MnFe(CN) ₆ after few cycles)	8
	NaK	1.0 M KClO ₄ in PC	3.5		~125		
K _{0.220} Fe[Fe(CN) ₆] _{0.805}	NaK	0.8 M KPF ₆ in EC/DEC (1:1 vol%)	3.2	114	107	Prussian blue analogue as host for K ions	9
Na ₂ C ₆ O ₆	NaK	1.0 M NaClO ₄ in EC/DEC (1:1 vol%) with 10 vol% FEC	2.2	250	~130	Carbonyl-based organic host with negligible preference toward Na and K	10
Poly(hexaazatriptycene)	NaK	1.5 M KPF ₆ in DME	1.8	--	~180	Organic polymer host with preference to K ions	11
TiO ₂ embedded NPCTO	NaK	1.0 M NaClO ₄ in EC/DEC (1:1 vol%) with 5 wt% FEC	1.0	--	~300	Capacitive capacity dominated	12
Ga-In	NaK	1.0 M NaClO ₄ in DME/FEC (95:5 vol%)	0.5	--	~80	Liquid alloy host with preference to Na ions	13

a) Theoretical capacity calculated assuming full (de)intercalation of alkali cations.

Abbreviations: FSI: bis(fluorosulfonyl)imide; Pyr₁₃: *N*-methyl-*N*-propylpyrrolidinium; PC: propylene carbonate; EC: ethyl carbonate; DEC: diethyl carbonate; FEC: fluoroethylene carbonate; DME: 1,2-dimethoxyethane; NPCTO: nitrogen-doped porous carbon truncated octahedra sheets

These results have been provided in the Supplementary Information section of our revised manuscript (Supplementary Table S4).

Reference

1. Basol, B. M. *et al.* Brief review of cadmium telluride-based photovoltaic technologies. *J. of Photonics for Energy*, **4(1)**, 040996 (2014).
2. Akiba, M. *et al.* Vulcanization and crosslinking in elastomers. *Prog. Polym. Sci.* **22**, 475521 (1997).
3. Zweibel, K. The Impact of Tellurium Supply on Cadmium Telluride Photovoltaics. *Science* **328(5979)**, 699-701 (2010).
4. Snyder, G. J. *et al.* Lead telluride alloy thermoelectrics. *Materialstoday* **14(11)**, 526-532 (2011).
5. Joshi, G. *et al.* Enhanced Thermoelectric Figure-of-Merit in Nanostructured p-type Silicon Germanium Bulk Alloys. *Nano Lett.* **8(12)**, 4670-4674 (2008).
6. Chivers, T. & Laitinen, R. S. Tellurium: a maverick among the chalcogens. *Chem. Soc. Rev.* **44**, 1725-1739 (2015).
7. Passerini, S. *et al.* A cost and resource analysis of sodium-ion batteries. *Nat. Rev. Mater.* **3**, 18013 (2018).
8. Xue, L. G., Gao, H. C., Li, Y. T. & Goodenough, J. B. Cathode dependence of liquid-alloy Na-K anodes. *J. Am. Chem. Soc.* **140**, 3292–3298 (2018).
9. Zhang, L. Y., Xia, X. H., Zhong, Y., Xie, D., Liu, S. F., Wang, X. L. & Tu, J. P. Exploring self-healing liquid Na-K alloy for dendrite-free electrochemical energy storage. *Adv. Mater.* **30**, 1804011 (2018).
10. Ding, Y., Guo, X. L. Qian, Y. M. Zhang, L. Y. Xue, L. G., Goodenough, J. B. & Yu, G. H. A liquid-metal-enabled versatile organic alkali-ion battery. *Adv. Mater.* **31**, 1806956 (2019).

11. Kapaev, R. R., Obrezkov, F. A., Stevenson, K. J. & Troshin, P. A. Metal-ion batteries meet supercapacitors: high capacity and high rate capability rechargeable batteries with organic cathode and a Na/K alloy anode. *Chem. Commun.* **55**, 11758–11761 (2019).
12. Huang, M., Xi, B. J., Feng, Z. Y., Wu, F. F., Wei, D. H., Liu, J. Feng, J. K., Qian, Y. T. & Xiong, S. L. New insights into the electrochemistry superiority of liquid Na-K alloy in metal batteries. *Small* **15**, 1804916 (2019).
13. Ding, Y., Guo, X. L., Qian, Y. M., Xue, L. G., Dolocan, A. & Yu, G. H. Room-temperature all-liquid-metal batteries based on fusible alloys with regulated interfacial chemistry and wetting. *Adv. Mater.* **32**, 2002577 (2020).

We believe that the fundamental understanding gained from this work can also be utilised to further design new high performance cathode materials for rechargeable NaK batteries. We have highlighted this point in the **DISCUSSION** section (page 19, line 26-line 32) as follows:

“With the use of non-terrigenous element such as Te, this study represents more of a fundamental interest. Nonetheless, related materials can be made useful for practical applications provided a terrigenous element can be found to replace Te in related mixed-alkali layered oxides.”

3) Techniques exist for analysing the XRD data - either by performing a Diffax simulation or more detailed Rietveld analysis using the FAULTS program. One or other of these methods should be employed here.

Response: To assuage the reviewer’s concern, we shall hereafter give a systematic account of the approaches taken to unravel the structural intricacies of NaKNi₂TeO₆ mixed-alkali honeycomb layered oxides. **Figure R2-3** shows the conventional and synchrotron X-ray diffraction (XRD) patterns of a pure-phase NaKNi₂TeO₆ prepared via a high-temperature solid-state reaction of stoichiometric mixture of Na₂Ni₂TeO₆ and K₂Ni₂TeO₆. Detail relating to the measurement protocols have been furnished in the **METHODS** section.

Figure R2-3. Conventional XRD pattern of $\text{NaKNi}_2\text{TeO}_6$ prepared via high-temperature solid-state reaction of $\text{Na}_2\text{Ni}_2\text{TeO}_6$ and $\text{K}_2\text{Ni}_2\text{TeO}_6$. (a) Conventional XRD patterns reveal a highly crystalline sample, although some diffraction peaks are highly asymmetric and broad. Schematic showing the synthesis protocol is shown (in inset). (b) Rietveld refinement plots of the synchrotron XRD pattern of $\text{NaKNi}_2\text{TeO}_6$ indexed in a hexagonal cell ($P\bar{6}2m$ space group model (shown in inset and also in Figure R2-4a)). Although most diffraction peaks are indexed using this model, a Bragg peak centered at 7.86° (shown in asterisk) is not fitted and the

pattern appears not well matched. The refined lattice parameters are: $a = 5.226(1)$ Å and $c = 11.7840(4)$ Å. The isotropic thermal parameters (U_{iso}) of the K1, K2, K3 sites were constrained. The z coordinates and U_{iso} of Ni1, Te1, Ni2, Te2 were also constrained. The reliability factors attained were as follows: $R_{\text{wp}} = 13.84\%$, $R_{\text{p}} = 9.57\%$, $\text{GOF} = 6.37$, warranting more analyses based on other structural models.

These results have been provided in the Supplementary Information section of our revised manuscript (Supplementary Figure S9).

Conventional XRD patterns (**Figure R2-3a**) indicate a highly crystalline $\text{NaKNi}_2\text{TeO}_6$ sample, although peak asymmetry and broadening was observed. Synchrotron XRD measurements were performed in order to conduct precise structural analyses, refinement data of which is shown in **Figure R2-3b**. A hexagonal cell was adopted (indexed in $P\bar{6}2m$ space group) was used as an initial model to depict a honeycomb layered framework with (i) an alternating arrangement of Na and K, and (ii) a disordered arrangement of Te and Ni atoms within the slabs, as shown in inset of **Figures R2-3a** and **R2-3b**. However, we faced a challenge of full indexation of the Bragg peaks (for instance, the peak centered around 8°) as highlighted in asterisk in **Figure R2-3b**.

Although another model indexed in $P\bar{6}2c$ hexagonal space group (with double periodicity) was used (shown in **Figure R2-4b**) in an attempt to fully fit the Bragg peaks of $\text{NaKNi}_2\text{TeO}_6$, the reliability values attained did not warrant the best fitting. **Figure R2-5** shows the Rietveld refinement plots of $\text{NaKNi}_2\text{TeO}_6$ indexed in $P\bar{6}2c$ hexagonal space group model (shown in **Figure R2-4b**). **Table R2-2** shows the refined unit cell parameters and atomic coordinates. Rietveld refinement of $\text{NaKNi}_2\text{TeO}_6$ using a larger unit cell led to poorer agreement values of $R_{\text{wp}} = 13.84\%$, $R_{\text{p}} = 9.57\%$, and goodness-of-fit (GOF) = 6.37. Several issues can be seen everywhere in the refinement and simply arise from the stacking disorder of the $[\text{Ni}_2\text{Te}]$ layers separated with Na.

To ascertain the appropriate model to describe the structure of $\text{NaKNi}_2\text{TeO}_6$ and unravel other structural intricacies (for instance, the presence of stacking disorders/faults, *et cetera*), atomic-resolution imaging was conducted on pristine $\text{NaKNi}_2\text{TeO}_6$ along multiple zone axes using spherical aberration-corrected scanning transmission electron

microscopy (Cs-corrected STEM). Information relating to the sample preparation, measurement protocols and the caveats undertaken are explicated in the **METHODS** section.

Figure R2-4. Structural models (hexagonal space groups) used to index the XRD pattern of NaKNi₂TeO₆. (a) A hexagonal cell ($P\bar{6}2m$) model used to initially index the synchrotron XRD data of NaKNi₂TeO₆ shown in Figure 3b. (b) A model with double periodicity (indexed in $P\bar{6}2c$ space group).

These results have been provided in the Supplementary Information section of our revised manuscript (Supplementary Figure S10).

Figure R2-5. Rietveld refinement plots of the synchrotron XRD pattern of $\text{NaKNi}_2\text{TeO}_6$ indexed in a hexagonal cell ($P\bar{6}2c$ space group model (shown in Figure 4b)). The reliability factors attained were as follows: $R_{\text{wp}} = 13.84\%$, $R_p = 9.57\%$ and $\text{GOF} = 6.37$, warranting more analyses based on transmission electron microscopy. (Supplementary Figure S15)

Table R2-2. Structural parameters obtained from the Rietveld refinement of the synchrotron XRD pattern of $\text{NaKNi}_2\text{TeO}_6$ in the $P\bar{6}2c$ hexagonal space group

Atom	Occupancy	x	y	z	U_{iso}
Te1	1	2/3	1/3	0.11848(11)	0.0077(4)
Ni1	1	0	0	0.1193(3)	0.003
Ni2	1	1/3	2/3	0.11681(18)	0.003
K1	1/3	0	0	1/4	0.105(7)
K2	1/3	2/3	0	1/4	0.105(7)
K3	1/3	2/3	1/3	1/4	0.105(7)
K4	1/3	1/3	2/3	1/4	0.105(7)
Na1	2/3	1/3	0	0	0.046(5)
O1	1	1/3	1/3	0.0689(5)	0.033(3)
O2	1	1/3	0	0.1602(3)	0.0003(14)

$P\bar{6}2c$ hexagonal space group with lattice parameters: $a = 5.22524(12)$ Å and $c = 23.561(1)$ Å. $R_{wp} = 13.84\%$, $R_p = 9.57\%$, GOF = 6.37. The isotropic thermal factors (U_{iso}) of the K1, K2, K3 and K4 sites were constrained. The U_{iso} of Ni1 and Ni2 were constrained to 0.003 Å². (Supplementary Table S2)

Figure R2-6. Visualisation of the atomic structure of NaKNi₂TeO₆ along the [100] and the [110] zone axes based on atomic-resolution TEM. (a) HAADF-STEM image taken along the [100] zone axis, depicting the atomic arrangement of Te and Ni modelled in a double periodicity cell akin to that shown in Figure 4b. (b)

Corresponding ABF-STEM image depicting an alternating arrangement of Na and K in the model that is in accord with the intensity distribution of the atom spots observed in the image. (c) HAADF-STEM images taken along the $[1\bar{1}0]$ and (d) Corresponding ABF-STEM images affirming the arrangement of atoms in $\text{NaKNi}_2\text{TeO}_6$ as depicted in (a) and (b).

These results have been provided in the revised manuscript (Figures 3b, 3c, 3g and 3h).

Figure R2-6 above shows the HAADF-STEM and ABF-STEM images taken along the $[100]$ and $[1\bar{1}0]$ axes, wherein the TEM atomic model matches well with the hexagonal lattice model having a double periodicity (indexed in a $P\bar{6}2c$ space group). Comparison of the experimentally obtained SAED patterns with various structural models (Figure R2-7) further ascertain the $P\bar{6}2c$ as the most appropriate space group model to index the structure of $\text{NaKNi}_2\text{TeO}_6$.

Figure R2-7. Comparison of the selected area electron diffraction (SAED) patterns of $\text{NaKNi}_2\text{TeO}_6$ experimentally taken with those of the proposed structural models, indicating the $P\bar{6}2c$ double periodicity hexagonal model as the most appropriate model to describe the structure of $\text{NaKNi}_2\text{TeO}_6$.

These results have been provided in the Supplementary Information section of our revised manuscript (Supplementary Figure S14).

Structural intricacies/disorders of the mixed-alkali honeycomb layered oxide $\text{NaKNi}_2\text{TeO}_6$ are further divulged by atomic-resolution STEM images (**Figure R2-8**), which reveal the existence of stacking faults wherein $\pm 1/3$ shifts of the Te / Ni slabs are observed (indicated by green arrows).

Figure R2-8. Direct visualisation of stacking defects along the c -axis [001] direction. (a) Low magnification ABF-STEM images showing streaks indicative of planar defects in a crystallite of $\text{NaKNi}_2\text{TeO}_6$. (b) High magnification HAADF-STEM images showing shifts in the Te/Ni slab at regions where the stacking faults are found to occur in $\text{NaKNi}_2\text{TeO}_6$. FAULTS program was further employed to quantitatively assess the nature of stacking faults inherent in $\text{NaKNi}_2\text{TeO}_6$, as detailed in the discussion below.

These results have been provided in the Supplementary Information section of our revised manuscript (Supplementary Figure S17).

To further quantitatively scrutinise the nature of the stacking faults innate in $\text{NaKNi}_2\text{TeO}_6$ as revealed by TEM (in **Figure R2-8**), FAULTS program¹ was employed. The atomic structural model indexed in the $P\bar{6}2c$ hexagonal space group was used as the initial model to perform the analyses of the stacking faults using the FAULTS program. **Figure R2-9** shows the refined SXRD pattern of $\text{NaKNi}_2\text{TeO}_6$ that was utilised to perform analyses

based on the FAULTS program. Nevertheless, good reliability factors were obtained (with GOF (χ^2) = 1.604). Various shift vectors were adopted to describe the stacking faults in NaKNi₂TeO₆ lattice along the *c* plane, amongst which, shift vectors having magnitudes of $[-1/3, -1/3, 0]$, $[1/3, 0, 0]$ and $[0, 1/3, 0]$ were found to be dominant with stacking probabilities of 7.5%, 6.3% and 4.1%, respectively. Taking all the results altogether, NaKNi₂TeO₆ possesses a high degree of stacking faults and we anticipate further study on the defect chemistry and physics of this class of honeycomb layered oxide materials in another scope of work.

Figure R2-9. Refined synchrotron XRD pattern of NaKNi₂TeO₆, used to perform quantitative analyses of stacking faults with the FAULTS¹ program. An enlarged image of the low-angle Bragg diffraction peaks is highlighted. The final lattice parameters obtained are as follows: $a = 5.2269(2)$ Å and $c = 23.5962(2)$ Å with a goodness-of-fit (GOF) value of 1.604. Stacking vectors having a magnitude of $[-1/3, -1/3, 0]$, $[1/3, 0, 0]$ and $[0, 1/3, 0]$ were found to be dominant with stacking probabilities of 7.5%, 6.3% and 4.1%, respectively. These results have been provided in the Supplementary Information section of our revised manuscript (Supplementary Figure S18).

Reference

1. Casas-Cabanas, M. *et al.* FAULTS: a program for refinement of structures with extended defects. *J. Appl. Cryst.*, **49**, 2259-2269 (2016).

4) The electrochemistry measurements are interesting, but why use tungsten foil for the current collector? The density surely reduces the precision of measuring the active mass of material.

Response: Owing to the relatively high operation voltage of this material, the upper cut-off voltage for Na, K and NaK half-cells were set at 4.35 V, 4.60 V and 4.35 V, respectively. Tungsten current collectors were therefore used instead of Al in order to eliminate possible oxidative corrosion of the current collector at high-voltage regimes. Besides, we are cognisant that the high density of tungsten metal foil may reduce the precision of measuring the mass of the active materials. We affirm that the weight measurement error caused by the use of tungsten current collectors is negligible as an ultra-micro balance with a 0.001 mg readability was used (versus ~5 mg of active material).

5) Also, the voltage limits used are particularly wide - the lower cut-off is unnecessarily low.

Response: With respect to the lower cut-off voltage, it is worth pointing out that decreasing the lower cut-off voltage to 1.20 V is found to favour more complete insertion of alkali ions into the layered structure, resulting in improved reversibility as can be seen in **Figure R2-10** below.

Figure R2-10. Voltage–capacity profiles of $\text{NaKNi}_2\text{TeO}_6$ cathode with a Na anode and 0.5 M NaFSI + 0.5 M KFSI in $\text{Pyrr}_{13}\text{FSI}$ electrolyte in two cut-off voltage ranges measured at C/20 (6.65 mA g^{-1}) and 25°C .

6) The authors make much of this idea of the mixed alkali dendrite free battery system, but then proceed to bury the data in the supplementary which show that it does not actually work as hoped.

Response: We thank the reviewer for giving us a chance to clarify this point. We have performed additional electrochemical measurements to justify $\text{NaKNi}_2\text{TeO}_6$ to be electrochemically active in NaK half-cells. The electrochemical data have been provided in **Figure 5** of the revised manuscript. Further optimisation of the electrode is subject of future work.

7) Finally, the language used in parts of the manuscript is unnecessarily elaborate.

Response: We have strived to ensure that the language is succinct, mindful that *Nature Communications* is committed to publishing important advances of significance to a multidisciplinary research across fields. Terminologies of a certain field may have different connotations to other fields, so we have ensured that we substantiate elaborately, in some

cases, to avoid any confusion to such readers. We therefore feel, *sensu lato*, that it is not anathema to be elaborate in some sentences as terminologies may have different connotations.

COMMENTS FROM REFEREE #3:

This paper reports on the mixed-alkali atom-based honeycomb layered oxide (NaK)Ni₂TeO₆ and its characterization using atomic-resolution STEM analysis. The authors reveal interesting local atomic structural disorders of aperiodic stacking and incoherency in the alternating arrangement of Na and K atoms. Specifically, (NaK)Ni₂TeO₆ shows an intriguing alternating arrangement of Na and K atoms sandwiched between Ni_{2/3}Te_{1/3} layers, which was exclusively probed by STEM analysis. This result provides an opportunity to tune the functionality of honeycomb layered oxides by controlling mixed alkali atoms in many applications. However, unlike the STEM analysis, XRD analysis is somehow shallow and needs a more precise and quantitative analysis. Therefore, I cannot recommend its publication in the present form.

We sincerely thank the reviewer for providing constructive remarks and valuable comments, which were of great help in revising the manuscript. Accordingly, the revised manuscript has been systematically improved with lucidity of discussion.

Major comments:

The following issues should be addressed before its publication;

1) While the STEM analysis is excellently executed, the X-ray diffraction analysis is not quantitative and somehow shallow. The authors demonstrated an atomic model structure that accurately describes the observed atomic arrangement by STEM in the atomistic range. However, they failed to describe the long-range crystal structure of (NaK)Ni₂TeO₆ based on the XRD result. As mentioned by the authors (page 6; line 156), newly emerged peaks in diffraction patterns for (NaK)Ni₂TeO₆ are forbidden in the *P6₃/mcm* symmetry. These new peaks should be emerged by the alternating arrangement of Na and K atoms along the *c*-direction with an X-ray coherency. Calculation of lattice parameters via profile matching is rather naive. The authors should resolve the detailed crystal structure (long-range) by using their proposed atomistic model, confirmed by the Rietveld refinement. Neutron diffraction analysis is also highly recommended.

Response: Thank you greatly for the comment. To allay the reviewer's concerns, we provide below an account of the various methodologies used to resolve the detailed crystal structure of mixed-alkali honeycomb layered oxide $\text{NaKNi}_2\text{TeO}_6$. Since the scattering length (form factor) of Na and K is essentially the same with neutron diffraction (*viz.*, 3.630 and 3.670, respectively), it renders it difficult to distinguish the Na and K atoms with neutron diffraction (data of which is appended in the discussion herein). We have therefore used mainly synchrotron XRD complementing STEM measurements to perform the structural analyses.

Figure R3-1 shows the conventional and synchrotron X-ray diffraction (XRD) patterns of a pure-phase $\text{NaKNi}_2\text{TeO}_6$ prepared via a high-temperature solid-state reaction of stoichiometric mixture of $\text{Na}_2\text{Ni}_2\text{TeO}_6$ and $\text{K}_2\text{Ni}_2\text{TeO}_6$. Detail relating to the measurement protocols have been furnished in the **METHODS** section.

Conventional XRD patterns (**Figure R3-1a**) indicate a highly crystalline $\text{NaKNi}_2\text{TeO}_6$ sample, although peak asymmetry and broadening was observed. Synchrotron XRD measurements were performed in order to conduct precise structural analyses, refinement data of which is shown in **Figure R3-1b**. A hexagonal cell was adopted (indexed in $P\bar{6}2m$ space group) was used as an initial model to depict a honeycomb layered framework with (i) an alternating arrangement of Na and K, and (ii) a disordered arrangement of Te and Ni atoms within the slabs, as shown in inset of **Figures R3-1a** and **R3-1b**. However, we faced a challenge of full indexation of the Bragg peaks (for instance, the peak centered around 8°) as highlighted in asterisk in **Figure R3-1b**.

Figure R3-1. Conventional XRD pattern of $\text{NaKNi}_2\text{TeO}_6$ prepared via high-temperature solid-state reaction of $\text{Na}_2\text{Ni}_2\text{TeO}_6$ and $\text{K}_2\text{Ni}_2\text{TeO}_6$. (a) Conventional XRD patterns reveal a highly crystalline sample, although some diffraction peaks are highly asymmetric and broad. Schematic showing the synthesis protocol is shown (in inset). (b) Rietveld refinement plots of the synchrotron XRD pattern of $\text{NaKNi}_2\text{TeO}_6$ indexed in a hexagonal cell ($P\bar{6}2m$ space group model (shown in inset)). Although most diffraction peaks are indexed using this model, a Bragg peak centered at 7.86° (shown in asterisk) is not fitted and the pattern appears not well

matched. The refined lattice parameters are: $a = 5.226(1)$ Å and $c = 11.7840(4)$ Å. The isotropic thermal parameters (U_{iso}) of the K1, K2, K3 sites were constrained. The z coordinates and U_{iso} of Ni1, Te1, Ni2, Te2 were also constrained. The reliability factors attained were as follows: $R_{\text{wp}} = 11.56\%$, $R_p = 8.22\%$ and goodness-of-fit (GOF) = 5.31, warranting more analyses based on other structural models.

These results have been provided in the Supplementary Information section of our revised manuscript (Supplementary Figure S9).

Although another model indexed in $P\bar{6}2c$ hexagonal space group (with double periodicity) was used (shown in **Figure R3-2b**) in an attempt to fully fit the Bragg peaks of $\text{NaKNi}_2\text{TeO}_6$, the reliability values attained did not warrant the best fitting. **Figure R3-3** below shows the Rietveld refinement plots of $\text{NaKNi}_2\text{TeO}_6$ indexed in $P\bar{6}2c$ hexagonal space group model (shown in **Figure R3-2b**). **Table R3-1** shows the refined unit cell parameters and atomic coordinates. Rietveld refinement of $\text{NaKNi}_2\text{TeO}_6$ using a larger unit cell (double-periodicity model) led to poorer agreement values of $R_{\text{wp}} = 13.84\%$, $R_p = 9.57\%$, and GOF = 6.37. Several issues can be seen everywhere in the refinement and simply arise from the stacking disorder of the $[\text{Ni}_2\text{Te}]$ layers separated with Na.

Rietveld refinement of neutron diffraction (ND) data was also performed using the hexagonal lattice model indexed in $P\bar{6}2c$ space group (**Figure R3-2b**). The refined lattice parameters are in good agreement with those obtained from refinement of the high-resolution synchrotron XRD data: $a = 5.227237(8)$ Å and $c = 23.58788(6)$ Å and the agreement parameters were as follows: $R_{\text{wp}} = 10.76\%$, $R_p = 8.92\%$ and GOF = 6.79. The refinement plots are shown as **Figure R3-4** whilst the refined atomic coordinates are provided in **Table R3-2**. However, several issues with peak intensity and asymmetric peak profile, which arise from the stacking disorder of the $[\text{Ni}_2\text{Te}]$ layers, can be seen in the ND refinement as well.

Figure R3-2. Structural models (hexagonal space groups) used to index the XRD pattern of $\text{NaKNi}_2\text{TeO}_6$. (a) A hexagonal cell ($P\bar{3}12$) model used to initially index the synchrotron XRD data of $\text{NaKNi}_2\text{TeO}_6$ shown in Figure 3b. (b) A model with double periodicity (indexed in $P\bar{6}2c$ space group).

These results have been provided in the Supplementary Information section of our revised manuscript (Supplementary Figure S10).

Figure R3-3. Rietveld refinement plots of the synchrotron XRD pattern of $\text{NaKNi}_2\text{TeO}_6$ indexed in a hexagonal cell ($P\bar{6}2c$ space group model (shown in Figure 4b)). The reliability factors attained were as follows: $R_{wp} = 13.84\%$, $R_p = 9.57\%$ and $GOF = 6.37$, warranting more analyses based on transmission electron microscopy. (Supplementary Figure S15)

Table R3-1. Structural parameters obtained from the Rietveld refinement of $\text{NaKNi}_2\text{TeO}_6$ in the $P\bar{6}2c$ hexagonal space group.

Atom	Occupancy	x	y	z	U_{iso}
Te1	1	2/3	1/3	0.11848(11)	0.0077(4)
Ni1	1	0	0	0.1193(3)	0.003
Ni2	1	1/3	2/3	0.11681(18)	0.003
K1	1/3	0	0	1/4	0.105(7)
K2	1/3	2/3	0	1/4	0.105(7)
K3	1/3	2/3	1/3	1/4	0.105(7)
K4	1/3	1/3	2/3	1/4	0.105(7)
Na1	2/3	1/3	0	0	0.046(5)
O1	1	1/3	1/3	0.0689(5)	0.033(3)
O2	1	1/3	0	0.1602(3)	0.0003(14)

$P\bar{6}2c$ hexagonal space group with lattice parameters: $a = 5.22524(12)$ Å and $c = 23.561(1)$ Å. $R_{wp} = 13.84\%$, $R_p = 9.57\%$, $GOF = 6.37$. The U_{iso} of the K1, K2, K3 and K4 sites were

constrained. The U_{iso} of Ni1 and Ni2 were constrained to 0.003 \AA^2 . (Supplementary Table S2)

Figure R3-4. Rietveld refinement plots of the neutron diffraction (ND) pattern of $\text{NaKNi}_2\text{TeO}_6$ indexed in a hexagonal cell ($P\bar{6}2c$ space group model (shown in Figure 2b)). Peaks that could not be well-fitted in the following d ranges were excluded from the refinement: $1.513\text{--}1.522 \text{ \AA}$, $1.614\text{--}1.636 \text{ \AA}$, $1.990\text{--}2.016 \text{ \AA}$, $2.200\text{--}2.228 \text{ \AA}$, and $2.560\text{--}2.578 \text{ \AA}$. Several issues with peak intensity and asymmetric peak profile as noted in the SXRD data, which arise from the stacking disorder of the $[\text{Ni}_2\text{Te}]$ layers, are apparent also in the ND refinement. The reliability factors attained were as follows: $R_{wp} = 10.76\%$, $R_p = 8.92\%$, $GOF = 6.78$.

These results have been provided in the Supplementary Information section of our revised manuscript (Supplementary Figure S16).

Table R3-2. Structural parameters obtained from the Rietveld refinement of $\text{NaKNi}_2\text{TeO}_6$ in the $P\bar{6}2c$ hexagonal space group.

Atom	Occupancy	x	y	z	$B_{\text{iso}} (\text{\AA}^2)$
Te1	1	2/3	1/3	0.117611(7)	0.231(3)
Ni1	1	0	0	0.117611(7)	0.231(3)
Ni2	1	1/3	2/3	0.117611(7)	0.231(3)
K1	1/3	0	0	1/4	0.231(3)
K2	1/3	2/3	0	1/4	0.231(3)
K3	1/3	2/3	1/3	1/4	0.231(3)
K4	1/3	1/3	2/3	1/4	0.231(3)
Na1	2/3	1/3	0	0	0.231(3)
O1	1	1/3	1/3	0.071957(13)	0.614(3)
O2	1	1/3	0	0.161895(13)	0.614(3)

$P\bar{6}2c$ hexagonal space group with lattice parameters: $a = 5.227237(8)$ \AA and $c = 23.58788(6)$ \AA. The attained reliability factors are: $R_{\text{wp}} = 10.76\%$, $R_{\text{p}} = 8.92\%$, $\text{GOF} = 6.78$. Note that the z coordinate of the Te1, Ni1, and Ni2 were constrained. The isotropic thermal factors (B_{iso}) of all cations were constrained. The B_{iso} values of O1 and O2 were constrained. (Supplementary Table S3)

To ascertain the appropriate model to describe the structure of $\text{NaKNi}_2\text{TeO}_6$ and unravel other structural intricacies (for instance, the presence of stacking disorders/faults, *et cetera*), atomic-resolution imaging was conducted on pristine $\text{NaKNi}_2\text{TeO}_6$ along multiple zone axes using spherical aberration-corrected scanning transmission electron microscopy (Cs-corrected STEM). Information relating to the sample preparation, measurement protocols and the caveats undertaken are explicated in the **METHODS** section.

Figure R3-5 below shows the HAADF-STEM and ABF-STEM images taken along the $[100]$ and $[1\bar{1}0]$ zone axes, wherein the TEM atomic model matches well with the hexagonal lattice model having a double periodicity (indexed in a $P\bar{6}2c$ space group). Comparison of the experimentally obtained SAED patterns with various structural models

(Figure R3-6) further ascertain the $P\bar{6}2c$ as the most appropriate space group model to index the structure of $\text{NaKNi}_2\text{TeO}_6$.

Figure R3-5. Visualisation of the atomic structure of $\text{NaKNi}_2\text{TeO}_6$ along the $[100]$ and the $[1\bar{1}0]$ zone axes based on atomic-resolution TEM. (a) HAADF-STEM image taken along the $[100]$ zone axis, depicting the atomic arrangement of Te and Ni modelled in a double periodicity cell akin to that shown in Figure R3-2b. (b) Corresponding ABF-STEM image depicting an alternating arrangement of Na and K in the model that is in accord with the intensity distribution of the atom spots

observed in the image. (c) HAADF-STEM images taken along the $[1\bar{1}0]$ and (d) Corresponding ABF-STEM images affirming the arrangement of atoms in $\text{NaKNi}_2\text{TeO}_6$ as depicted in (a) and (b).

These results have been provided in the revised manuscript (Figures 3b, 3c, 3g and 3h).

Figure R3-6. Comparison of the selected area electron diffraction (SAED) patterns of $\text{NaKNi}_2\text{TeO}_6$ experimentally taken with those of the proposed structural models, indicating the $P\bar{6}2c$ double periodicity hexagonal model as the most appropriate model to describe the structure of $\text{NaKNi}_2\text{TeO}_6$.

These results have been provided in the Supplementary Information section of our revised manuscript (Supplementary Figure S14).

2) While only one report on a set of metastable antimonates $\text{Li}_{3-x}\text{Na}_x\text{Ni}_2\text{SbO}_6$ has been published (ref 24), the present mixed-alkali atom honeycomb layered oxides ($\text{Na}_{2-x}\text{K}_x\text{Ni}_2\text{TeO}_6$) seems stable. What is the origin of this structural stability? Can the authors explain from the thermodynamic point of view? DFT calculation would be very helpful to explain.

Response: Thank you greatly for the comment. Reference 24 investigates the intermediate honeycomb layered antimonate compositions between O3-type $\text{Na}_3\text{Ni}_2\text{SbO}_6$ ($\text{NaNi}_{2/3}\text{Sb}_{1/3}\text{O}_2$) and O3-type $\text{Li}_3\text{Ni}_2\text{SbO}_6$ ($\text{LiNi}_{2/3}\text{Sb}_{1/3}\text{O}_2$), wherein they design a mixed alkali honeycomb layered oxide $\text{Li}_{3-x}\text{Na}_x\text{Ni}_2\text{SbO}_6$ ($x = 1.5$) with a random occupancy distribution of Li and Na. The structure however decomposes to the parent $\text{Na}_3\text{Ni}_2\text{SbO}_6$ and $\text{Li}_3\text{Ni}_2\text{SbO}_6$ phases upon reannealing, revealing its metastability. Theoretical computations indicate an increase in the phase formation energies of various configuration of $\text{Li}_{3-x}\text{Na}_x\text{Ni}_2\text{SbO}_6$ relative to the decomposed products (*i.e.*, $\text{Na}_3\text{Ni}_2\text{SbO}_6$ and $\text{Li}_3\text{Ni}_2\text{SbO}_6$), affirming the conclusions drawn from the work (reference 24).

(i) **Alternating arrangement of Na and K in $\text{NaKNi}_2\text{TeO}_6$ and its stability.**

In our pursuit to design mixed alkali honeycomb layered tellurates, the intermediate structural composition of P2-type $\text{K}_2\text{Ni}_2\text{TeO}_6$ and P2-type $\text{Na}_2\text{Ni}_2\text{TeO}_6$ display not a randomly mixed but an alternating arrangement of Na and K in the intermediate mixed alkali composition $\text{NaKNi}_2\text{TeO}_6$. The alternating arrangement of Na and K in the structure of $\text{NaKNi}_2\text{TeO}_6$ was affirmed by atomic-resolution imaging on multiple zone axes conducted using spherical aberration-corrected scanning transmission electron microscopy (Cs-corrected STEM), as is shown in **Figure R3-7** below. The structural stability of $\text{NaKNi}_2\text{TeO}_6$ upon reannealing was validated by XRD measurements (shown in **Figure R3-8**), which indicate similarity in the XRD patterns (before and after annealing) thereby affirming mixed alkali framework of $\text{NaKNi}_2\text{TeO}_6$ to be stable.

Figure R3-7. Atomic-resolution ABF-STEM imaging of $\text{NaKNi}_2\text{TeO}_6$ indicating unambiguously the alternating arrangement of Na and K along the c -axis. These results have been provided in the revised manuscript (Figures 3c and 3h).

Figure R3-8. A comparison of the XRD patterns of as-prepared and reannealed $\text{NaKNi}_2\text{TeO}_6$, revealing structural stability of $\text{NaKNi}_2\text{TeO}_6$ upon thermal treatment. The wavelength was set at $\text{Cu } K\alpha$.

These results have been provided in the Supplementary Information section of our revised manuscript (Supplementary Figure S33).

(ii) Rationalising the origin of NaK₂TeO₆ stability.

It is reasonable to assume that the alternating arrangement of Na and K in separate layers in the structure of NaK₂TeO₆ guarantees its stability, since a randomly mixed Na and K configuration akin to that of Li_{1.5}Na_{1.5}Ni₂SbO₆ possesses higher configuration entropy.

Looking at the structure of NaK₂TeO₆, a prominent aspect that emerges is the difference between Na and K sites evident when NaK₂TeO₆ crystals are viewed along the [1 $\bar{1}$ 0] zone axes as shown in **Figure R3-7a**. Na atoms are distributed in sites that assume triangular patterns (**Figure R3-9a**) whilst the K atoms reside in sites arranged in hexagonal formations (**Figure R3-9b**). The atomic arrangements of the atoms given in **Figure R3-9** can be intricately linked to crystalline entropy considerations, explanation of which we give below.

Figure R3-9. Arrangement of the sites occupied by Na and K atoms in NaK₂TeO₆ as viewed from the [001] zone axis. (a) Arrangement of the Na atoms in triangular

sites (as highlighted in red) as viewed along the [001] zone axis. (b) Hexagonal arrangement of K atoms in their respective sites as depicted along the [001] zone axis. Note that full occupancy of the sites is shown for brevity.

These results have been provided in Figure 4 of our revised manuscript.

In particular, the free energy of the alkali atom layer is expected to be minimised (or equivalently, the entropy maximised) to achieve stability of the crystal especially when the Na or K atoms are arranged in a honeycomb fashion. This follows from the honeycomb conjecture, which states that the honeycomb lattice is the most efficient way of packing any two-dimensional (2D) surface with equal size unit cells of maximum area and minimum perimeter.¹ Thus, taking the free energy F to scale with the perimeter of the unit cells (honeycomb, triangular *etc.*), and the entropy to scale with the area, A , the lattice exhibited by K atoms in **Figure R3-9b** satisfies this conjecture by maintaining its honeycomb pattern whilst Na atoms in **Figure R3-9a** do not.

For the sake of rigour, we make a straightforward approximation for the free energy, F using the thermodynamics formula, $F = U - k_B T \ln KA$, where K is a constant related to the geometry of the surface (Gaussian curvature),² $S = k_B \ln KA$ is the entropy contribution to the free energy and $U \simeq -E_a$ is the internal potential energy corresponding to various binding energies of the alkali atoms (Na, K) to each other, whose leading term is taken to be their activation energy, E_a . Following this formula, the entire material comprising a series of such layers has to maximise entropy, even for the mixed alkali atom honeycomb layered oxides. As affirmed earlier, the highest entropy configuration leading to a minimised free energy and a stable crystalline structure is the one where both types of alkali atoms are arranged in a honeycomb fashion. The next favoured configuration is the one that allows for only one type of alkali atom to be in a honeycomb fashion, case in point being the configuration observed in $\text{NaKNi}_2\text{TeO}_6$ as displayed in **Figures R3-9a** and **R3-9b**. Moreover, since the activation energy of K is lower than that of Na,^{3,4} it is apparent, by setting $F = 0$, that the Na layer can still minimise its free energy by disrupting its honeycomb configuration into e.g. triangular patterns

shown in **Figure R3-9a**. Thus, since the arrangement of the alkali atoms correlates with the adjacent metal slab atoms (Te, Ni), these slabs shear in order to accommodate the disruption, as observed in **Figure R3-10**.

Figure R3-10. (a) HAADF-STEM image illustrating the unique stacking sequence in $\text{NaKNi}_2\text{TeO}_6$. In the layers where K atoms occupy the interlayer space, Te / Ni slabs are not shifted with respect to each other (marked by a green line). However, for layers where Na atoms reside, shifts of the Te / Ni slabs are observed in a random direction. The yellow and blue lines show shifts in different directions. Note the aperiodicity in the stacking sequence. (b) Enlarged view of the domain highlighted in (a), showing that the interlayer distance is contingent upon the alkali atom species (Na or K) sandwiched between the Te / Ni layers.

These results have been provided in Figures 2g and 2h of our revised manuscript.

A similar entropic and free energy argument can be made to account for sodium and potassium spontaneously intercalating between the oxide layers separately albeit adjacent to each other. This corresponds to new binding energy terms in the potential energy inversely proportional to the separation distance of atoms of the same type. The simplest term can be written as $U(d) \approx -E_a - \alpha/d^n$ where α is a positive constant, n a positive integer and d the separation distance of the atoms (e.g. for the binding energy

offsetting the Coulomb repulsion between two alkali atoms of the same type, $\alpha = q^2/4\pi\epsilon$ is the fine structure constant and $n = 1$, where q is the charge of the adjacent alkali atom of the same type and ϵ the permittivity across the distance, d). Thus, since a small separation distance, d further lowers the free energy of the material, the lowest stable free energy configuration is where $d = 0.62$ nm or $d = 0.55$ nm corresponding to the case of the pure atom KNi_2TeO_6 or $\text{NaNi}_2\text{TeO}_6$ respectively, where the distances are correlated with the ionic radii of the respective alkali atoms. Moreover, this strengthens the above argument for the disruption of the honeycomb pattern for Na in $\text{NaKNi}_2\text{TeO}_6$, since this new binding energy term is smaller for Na atoms compared to the case for K atoms, by virtue of a smaller d for Na atoms compared to K atoms. Finally, the next favoured configuration is given by, $d = (0.55 + 0.62)$ nm, corresponding to the mixed alkali atom $\text{NaKNi}_2\text{TeO}_6$ reported herein, as shown in **Figure R3-10b**.

The $\pm 1/3$ shifts of the Te / Ni slabs where Na atoms reside observed from TEM inspection (**Figure R3-10b**), presumably account for the change in the site occupation and thus endowing structural stability of $\text{NaKNi}_2\text{TeO}_6$. To glean more on the stability of $\text{NaKNi}_2\text{TeO}_6$, theoretical studies based on density functional theory (DFT) calculations were performed based on intermediate $\text{NaKNi}_2\text{TeO}_6$ configurations with varying degrees of adjacent slab shifts where Na atoms reside, some of which are shown in **Figure R3-11** below. For clarity, the structural stability of the mixed alkali honeycomb configurations was studied by calculating their formation energies based on DFT. An electronic structure calculation code implemented in the Vienna ab initio simulation package (VASP) using DFT, further details of which are explicated in the **METHODS** section. Moreover, the formation energies for structural configurations of the parent phases (*i.e.*, $\text{K}_2\text{Ni}_2\text{TeO}_6$ and $\text{Na}_2\text{Ni}_2\text{TeO}_6$) including those with shifts of the Te/Ni slabs (as observed in our previous works^{5,6}) were also calculated (shown in **Figure R3-12**) to rationalise their phase stability particularly during electrochemical alkali-ion extraction and reinsertion where they are likely to be formed. This discussion is made in the DISCUSSION section (pages 16 and 18).

Figure R3-11. Exemplar configurations of intermediate $\text{NaKNi}_2\text{TeO}_6$ compositions (with varying shifts of the Te/Ni slabs where Na atoms reside) assessed using theoretical computation. Their formation energies were calculated based on DFT calculations.

These results have been provided in the Supplementary Information section of our revised manuscript (Supplementary Figure S26).

Figure R3-12. Exemplar configurations of $\text{Na}_2\text{Ni}_2\text{TeO}_6$ and $\text{K}_2\text{Ni}_2\text{TeO}_6$ parent phase compositions (some with varying shifts of the Te/Ni slabs where Na atoms reside) assessed using theoretical computation. Their formation energies were calculated based on DFT calculations.

These results have been provided in the Supplementary Information section of our revised manuscript (Supplementary Figure S27).

A summary of the total formation energies calculated from prime configurations are shown in **Figure R3-13**. Looking at the structural stability of $\text{NaKNi}_2\text{TeO}_6$, the configuration experimentally attained (structure (6)) is the most stable amongst others calculated (structures (5), (3) and (2)). This result shows that the shifts of the Te/Ni metal slab layers aid to guarantee the stability of layers with Na atoms in the layered framework of $\text{NaKNi}_2\text{TeO}_6$.

Figure R3-13. Total formation energies for various configurations of $\text{NaKNi}_2\text{TeO}_6$ (1, 4, 7 and 8, shown in Figure 9) and parent phases (2, 3, 5 and 6, shown in Figure S10) based on DFT calculations. The experimentally attained configuration (structure 6) is the most stable amongst others calculated (for instance, structures 5, 3 and 2).

These results have been provided in the Supplementary Information section of our revised manuscript (Supplementary Figure S25).

As for $\text{Na}_2\text{Ni}_2\text{TeO}_6$, the structure with shifts of the Te/Ni metal layers (structure ⑧) presents a lower formation energy (hence more stable); which is indeed what has been observed in pristine $\text{Na}_2\text{Ni}_2\text{TeO}_6$ and also upon electrochemical cycling of $\text{NaKNi}_2\text{TeO}_6$ in Na and NaK half-cells, as is shown in Figure R3-14.

Figure R3-14. Structural changes during alkali-ion (de)insertion of $\text{NaKNi}_2\text{TeO}_6$ upon subsequent cycling in NaK half-cells. (a) HAADF-STEM images of $\text{NaKNi}_2\text{TeO}_6$ upon cycling in NaK battery cell taken along [100] zone axis. Na layers

entailing shifts of the adjacent slabs are shown in green arrow. (b) Intensity line profiles as highlighted in (a) showing equidistant interlayer spacings (0.55 nm) along the [001] axis revealing NaK₂Ni₂TeO₆ to preferentially insert Na atoms into the lattice. This is further quantitatively illustrated in (c). For clarity, the horizontal axis in (c) shows the layer numbers.

These results have been added to RESULTS section of our revised manuscript (page 25, revised Figures 7a, 7b and 7c).

Reference

1. Hales, T. C. The honeycomb conjecture. *Discrete. Computr. Geom.* **25**, 1–22 (2001).
2. Kanyolo, G. M. & Masese, T. An Idealised Approach of Geometry and Topology to the Diffusion of Cations in Honeycomb Layered Oxide Frameworks. *Sci. Rep.* **10**, 13284 (2020).
3. Kanyolo, G. M. *et al.* Honeycomb Layered Oxides: Structure, Energy Storage, Transport, Topology and Relevant Insights. *Chem. Soc. Rev.* **50**, 3990–4030 (2021).
4. Matsubara, N. *et al.* Magnetism and Ion Diffusion in Honeycomb Layered Oxide K₂Ni₂TeO₆: First Time Study by Muon Spin Rotation & Neutron Scattering. *Sci. Rep.* **10**, 18305 (2020).
5. Masese, T. *et al.*, Unveiling structural disorders in honeycomb layered oxide: Na₂Ni₂TeO₆. *Materialia.* **15**, 101003 (2021).
6. Masese, T. *et al.*, Topological Defects and Unique Stacking Disorders in Honeycomb Layered Oxide K₂Ni₂TeO₆ Nanomaterials: Implications for Rechargeable Batteries. *ACS Appl. Nano Mater.* **4**, 279–287 (2021).
7. Ding, Y., Guo, X. L. Qian, Y. M. Zhang, L. Y. Xue, L. G., Goodenough, J. B. & Yu, G. H. A liquid-metal-enabled versatile organic alkali-ion battery. *Adv. Mater.* **31**, 1806956 (2019).

3) (page 17; line 442) "This is generally ascribed to the fact that the number of cations participating in extraction and reinsertion would be diminished. As a way to enhance performance, the utilisation of an alloy such as NaK in the case of $\text{NaKNi}_2\text{TeO}_6$ would facilitate the participation of both cation species thus yield a realistic theoretical capacity, as demonstrated in Figure S6."

=> I do not agree with the author's claim that utilizing both cation species (here Na^+ and K^+) may yield a realistic theoretical capacity in mixed-alkali battery systems. It is well known that the extraction of whole Na and/or K will collapse the layers, leading to severe capacity fading. The authors need to tone down the statements about the feasibility of mixed-alkali batteries throughout the manuscript.

Response: Thank you greatly for the comment. Our assertion of the feasibility of mixed-alkali batteries is not hyperbolic, as affirmed by a number of recent reports on the subject.¹⁻⁴

The ability to use liquid NaK metal alloy provides an opportunity to access a majority of both Na and K in mixed-alkali honeycomb layered oxides such as $\text{NaKNi}_2\text{TeO}_6$. We accede with the reviewer that extraction of the entire alkali atoms from layered oxide materials will debilitate their structural integrity, leading to drastic capacity attenuation as has been noted in mainstream cathode materials for the conventional lithium-ion batteries. However, with bespoke electrode coating processes close-to-theoretical capacity values have been realised in layered oxides, such as $\text{LiNi}_{0.8}\text{Co}_{0.1}\text{Mn}_{0.1}\text{O}_2$ (dubbed as 'NCM811').^{5,6} In the same vein, we anticipate that practical capacities close to the theoretical capacities are attainable with an optimised electrode design. This is subject of future work.

References

1. Goodenough, J. B. *et int.* Liquid K-Na Alloy Anode Enables Dendrite-Free Potassium Batteries. *Adv. Mater.* **28**, 9608 (2016).
2. Goodenough, J. B. *et int.* Cathode Dependence of Liquid-Alloy Na–K Anodes. *J. Am. Chem. Soc.* **140**, 3292 (2018).

3. Xiong, S. *et al.* New Insights into the Electrochemistry Superiority of Liquid Na–K Alloy in Metal Batteries. *Small* **15**, 1804916 (2019).
4. Chen, C. -Y. *et al.* An Energy - Dense Solvent - Free Dual - Ion Battery. *Adv. Funct. Mater.* **30**, 2003557 (2020).
5. Li, X. *et al.* LiNbO₃-coated LiNi_{0.8}Co_{0.1}Mn_{0.1}O₂ cathode with high discharge capacity and rate performance for all-solid-state lithium battery. *J. Energy Chem.* **40**, 39-45 (2020).
6. Meng, K. *et al.* Improving the cycling performance of LiNi_{0.8}Co_{0.1}Mn_{0.1}O₂ by surface coating with Li₂TiO₃. *Electrochim. Acta* **211**, 822-831 (2016).

4) (Page 9; line 220) "Figure 2a shows a high-angle annular dark-field (HAADF)-STEM image of the parent compound K₂Ni₂TeO₆."

=> Figure 2a should be Figure S2a.

Response: We thank the reviewer's careful examination on our manuscripts. We have corrected the typographical error (page 4, line 23) and gone through the manuscript carefully for any related errors.

REVIEWER COMMENTS

Reviewer #1 (Remarks to the Author):

I appreciate authors' great effort in responding four referees' comments and revising the manuscript. Now the quality is good enough for publication in this journal.

Reviewer #2 (Remarks to the Author):

The authors have performed one of the most comprehensive rebuttals I have ever observed, including performing many new experiments. This is now a much stronger piece of work which can be published. I stand by my remarks about the voltage window - the reduction in performance by changing from 1.2 to 2.5V is minimal.

Reviewer #3 (Remarks to the Author):

I appreciate the author's efforts to perform the additional experiments and revise the manuscript. I acknowledge all of the issues raised have been well addressed. I recommend a publication in its present form.

Responses to Reviewers' Comments

We once again thank the reviewers for their time expended to evaluate our manuscript favourably and for their overall assessment of this novel work relating to the studied class of mixed-alkali honeycomb layered oxides.

Manuscript number: NCOMMS-20-38537-A

Title: Mixed Alkali-Ion Transport and Storage in Atomic-Disordered Honeycomb Layered Oxide $\text{NaKNi}_2\text{TeO}_6$

COMMENTS FROM REFEREE #1:

I appreciate authors' great effort in responding to the four referees' comments and revising the manuscript. Now the quality is good enough for publication in this journal.

We sincerely thank the reviewer for the positive comment and support.

COMMENTS FROM REFEREE #2:

The authors have performed one of the most comprehensive rebuttals I have ever observed, including performing many new experiments. This is now a much stronger piece of work which can be published. I stand by my remarks about the voltage window - the reduction in performance by changing from 1.2 to 2.5 V is minimal.

We greatly appreciate the reviewer's positive remark on our efforts to revise the manuscript and the insightful remarks on the cut-off voltage. In response, we have provided the voltage (dis)charge curves of $\text{NaKNi}_2\text{TeO}_6$ set at a lower cut-off voltage of 2.5 V in Na- and K-half cells in the **Supplementary Information**.

COMMENTS FROM REFEREE #3:

I appreciate the author's efforts to perform the additional experiments and revise the manuscript. I acknowledge all of the issues raised have been well addressed. I recommend a publication in its present form.

We are very thankful for the reviewer's positive assessment of this work.